



# Glaciohydraulic seismic tremors on an Alpine glacier

Fabian Lindner[1], Fabian Walter[1], Gabi Laske[2], and Florent Gimbert[3]

[1]Laboratory of Hydraulics, Hydrology and Glaciology (VAW), ETH Zürich, Zürich, Switzerland
[2]Institute of Geophysics and Planetary Physics, Scripps Institution of Oceanography, UC San Diego, La Jolla, USA
[3]Institut des Géosciences de l'Environnement, Université Grenoble Alpes, UMR CNRS 5001, Grenoble, France

**Correspondence:** Fabian Lindner (lindner@vaw.baug.ethz.ch)

**Abstract.** Hydraulic processes impact viscous and brittle ice deformation. Water-driven fracturing as well as turbulent water flow within and beneath glaciers radiate seismic waves which provide insights into otherwise hard-to-access englacial and subglacial environments. In this study, we analyze glaciohydraulic tremors recorded by four seismic arrays installed in different parts of Glacier de la Plaine Morte, Switzerland. Data were recorded during the 2016 melt season including the sudden subglacial drainage of an ice-marginal lake. Together with our seismic data, discharge, lake level, and ice flow measurements provide constraints on glacier hydraulics. We find that the tremors are generated by subglacial water flow, in moulins, and by icequake bursts. The dominating process can vary on sub-kilometer and sub-daily scales. Consistent with field observations, continuous source tracking via matched-field processing suggests a gradual upglacier progression of an efficient drainage system as the melt season progresses. The ice-marginal lake likely connects to this drainage system via hydrofracturing, which is indicated by sustained icequake signals emitted from the proximity of the lake basin and starting roughly 24 hours prior to the lake drainage. To estimate the hydraulics associated with the drainage, we use tremor-discharge scaling relationships. Our analysis suggests a pressurization of the subglacial environment at the drainage onset, followed by an increase in the hydraulic radii of the conduits and a subsequent decrease in the subglacial water pressure as the capacity of the drainage system increases. The pressurization is in phase with the drop in the lake level and its retrieved maximum coincides with ice-uplift measured via GPS. Our results highlight the use of cryo-seismology for monitoring glacier hydraulics.

## 1 Introduction

On high-melt glaciers, meltwater produced at the surface is routed through moulins and crevasses to the glacier bed. Subglacially, the water flows in a drainage system often described by the two end-member scenarios of distributed and channelized flow (Fountain and Walder, 1998; Cuffey and Paterson, 2010). During the melt season with increased meltwater input, the subglacial drainage system typically transitions from the distributed to a channelized system allowing for more efficient water evacuation (Fountain, 1993; Hock and Hooke, 1993; Bartholomew et al., 2010). In the case that the drainage system does not adapt fast enough to meltwater input, subglacial water pressures increase. Such a configuration is often encountered in the early melt season (Iken and Bindschadler, 1986; Werder et al., 2013). In addition, drainage events of glacier-dammed lakes can inject large volumes of water on short time scales exceeding the capacity of the subglacial conduits and causing a pressurization of the system (Roberts, 2005). By modulating the effective pressure at the glacier bed, glacier hydraulics play a key role in ice



flow dynamics (Iken and Bindschadler, 1986). For instance, observed accelerations of Greenland outlet glaciers are attributed to increased meltwater availability (Zwally et al., 2002; Bartholomew et al., 2010), though the exact mechanisms are still under
debate (Schoof, 2010).

    Different approaches have been used to probe the subglacial drainage system. Borehole studies (e.g. Andrews et al., 2014) provide time-series of subglacial water pressure, ground-penetrating radar (e.g. Stuart et al., 2003) and active seismics (e.g. Nolan and Echelmeyer, 1999) enable the investigation of englacial and subglacial material properties, and dye tracer experiments (e.g. Werder et al., 2009) yield insights on water pathways through and beneath glaciers. However, these approaches
are expensive and laborious or provide subsurface images at only a few instances in time or isolated point measurements. In contrast, cryo-seismology (Podolskiy and Walter, 2016; Aster and Winberry, 2017) requires less manpower and allows continuous monitoring as well as spatial insights. Recent studies show that various processes related to glacial hydraulics radiate seismic waves that in turn can be used to investigate these processes. Similar to river-induced seismic noise (Gimbert et al., 2014), subglacial discharge generates seismic tremors due to pressure-fluctuations in turbulent flow and by impact events dur-
ing bedload transport. Bartholomaus et al. (2015) show that these tremors serve as proxy for subglacial discharge and find that the tremors reveal decreasing transit times of the water through the glacier throughout the melt season. Building on their river application, Gimbert et al. (2016) establish a glacier framework which relates seismic power $P_{rel}$ to discharge $Q_{rel}$ (using an arbitrary reference scaling). This framework allows the discrimination between the following end members of the subglacial drainage regime derived from an analytical model: i) Discharge routing through pressure-gradient adjustment in conduits of
constant hydraulic radius implying $P_{rel} \propto Q_{rel}^{14/3}$. This configuration is expected in cases where the conduits do not adjust their hydraulic radii fast enough to accommodate discharge changes, as is expected in the early melt season (Gimbert et al., 2016). ii) Discharge routing through conduits of varying hydraulic radius under constant pressure-gradient implying $P_{rel} \propto Q_{rel}^{5/4}$. This situation is e.g. expected for conduits transitioning from filled to unfilled. These scaling relationships are valid for seismic waves generated by efficient flow in multiple conduits as long as the number of conduits and their positions do not change.
Gimbert et al. (2016) test their framework on data from a bedrock station next to Mendenhall Glacier, Alaska, and find that over weekly and longer time scales radius adjustment is the dominant mechanism, while pressure-gradient variability is significant over the course of hours to days. Another study concludes that multi-channel flow can be distinguished from single-channel flow from the frequency structure of the tremors (Vore et al., 2019).

    In addition to tremors originating subglacially, a number of studies report on tremors generated in moulins (Roeoesli et al.,
2014; Walter et al., 2015; Roeoesli et al., 2016; Aso et al., 2017). Roeoesli et al. (2016) observe moulin tremors generated by resonances in the water column producing a fundamental frequency signal with overtones. They use the signal to invert for the moulin aspect ratio and depth using a semi-open organ pipe model.

    Apart from the continuous tremor signal, glacier hydraulics may give rise to discrete fracturing events. Given that sufficient meltwater is available, hydrofracturing can extend existing fractures to the glacier bed (Van Der Veen, 1998). Evidence for
such events in combination with resonances in water-filled cavities is reported e.g. in Helmstetter et al. (2015) who analyzed the recordings of an accelerometer deployed on ice. In case of high englacial water pressures exceeding the ice-overburden pressure, hydraulic jacking of the ice can occur. Jacking accompanied by seismicity is e.g. reported during rapid drainage events





of supraglacial lakes (Das et al., 2008; Doyle et al., 2013). We also note that high occurrence rates of overlapping fracturing icequakes may result in sustained tremor-like fracturing events (Podolskiy et al., 2018; MacAyeal et al., 2018).

In this study, we analyze data from on-ice seismic stations deployed during the 2016 melt-season on Glacier de la Plaine Morte, Switzerland (Sect. 2). We show that both tremors and icequake activity are linked to glacial discharge which includes the outburst-flood of a glacier-dammed lake (Sect. 3). By investigating the source locations of the tremor signals as seen from different arrays, we are able to attribute the tremors to different glacier hydraulics processes and shed light on the temporal evolution of the latter (Sect. 4). Finally, we discuss our results in the light of tremor origin, time evolution of the drainage
system and drainage regime (Sect. 5) and draw our conclusions (Sect. 6).

## 2   Field site and instrumentation

Glacier de la Plaine Morte (Fig. 1) in the Swiss Alps is located along the border of the cantons Bern and Valais. With a surface area of approximately 7.4 km$^2$ of which 90% occupies the narrow elevation range between 2650 and 2800 m a.s.l., Glacier de la Plaine Morte is the largest plateau glacier in the European Alps. From this plateau, a small outlet glacier called Rätzligletscher
flows to the Bernese side to the north. Except for the north-dipping topography in this area, the glacier surface can be considered flat (the average slope is less than 4 degrees), which implies that ice flow is negligible (measured summer surface velocities are smaller than 1 cm d$^{-1}$). In most years, the equilibrium line altitude in the study region is either above or below the plateau elevation, inhibiting a clear separation in accumulation and ablation area. For this reason, the glacier is extremely sensitive to changes in the climatic forcing (Huss and others, 2013). The maximum ice thickness is around 200 m. More details on Glacier
de la Plaine Morte are available through GLAMOS (Glacier Monitoring Switzerland GLAMOS, 2018).

In recent years, the annual filling and subglacial drainage of an ice-dammed lake, Lac des Faverges (Fig. 1), at the south-eastern rim of the glacier was observed, which increases the risk of flooding the Simme Valley to the north. In 2016, the lake reached a volume of $\approx 2\times 10^6$ m$^3$ which was released within six days at the end of August. In addition, a smaller supraglacial lake at the southern rim (labeled "SL" in Fig. 1) formed in 2016 and drained prior to Lac des Faverges.

Our field campaign started in late April with the installation of an array consisting of five Lennartz LE3D/BH seismometers in shallow boreholes. Above 1 Hz, the sensors have a flat response to ground velocity and they were connected to Nanometrics Centaur digitizers logging data at 500 samples per second. At the end of July (day 212), we added a sixth sensor of the same type to this array. The data of this station were recorded by an Omnicrecs DATA-CUBE3 at 200 samples per second. The aperture of this array was 360 m. In mid-July on days 202 to 204, we installed three additional arrays, each consisting of 5 stations with an
aperture of 100 m. For each of these stations, we used a three-component 4.5 Hz geophone (PE-6/B manufactured by SENSOR Nederland) connected to an Omnirecs DATA-CUBE3 logging ground velocity at 400 samples per second. The geophones were installed in the snow pack and later on ice (for details see Lindner et al., 2018) while the digitizers stayed at the surface to retain GPS capability. In the following, consistent with the station names, we refer to our four arrays as A0 (stations PM01-PM06), A1 (PM11-PM15), A2 (PM21-PM25), and A3 (PM31-PM35). While A0 recordings are continuous (apart from gaps due to station maintenance), recordings from the other arrays suffer from occasional power outages and frequently exhibit gaps over





midnight of up to 26 minutes. A0, A1, and A2 stations recorded data through early September (day 250 to 252, respectively), A3 stations were dismantled on August 23 (day 236) due to a slushy snow layer at the glacier's surface.

In addition to the seismogenic ground motion, we surveyed the (low-frequency) glacier surface motion due to e.g. ice flow and glacier hydraulics at three locations using GPS units (Fig. 1a). Furthermore, we make use of the following time series: discharge in the Simme river to the north (measured ≈4 km from the terminus of Rätzligletscher), level of the outlet stream (≈1.5 km from the terminus of Rätzligletscher), and level of Lac des Faverges. Simme discharge is measured by Switzerland's Federal Office for the Environment, and the stream and lake level are provided through a monitoring program conducted by

the municipality of Lenk and the company Geopraevent.

## 3    Data and observations

### 3.1    Discharge

Over the course of the 2016 melt season, Lac des Faverges steadily filled (orange dashed line in Fig. 2a) and reached a maximum volume of approximately two million cubic meters of water at the end of August. In the evening of 27 August (day 240), a lake

drainage through the moulin marked in Fig. 1(a) initiated, and emptied the lake basin in approximately six days. The moulin routed the water to the subglacial environment and it escaped the glacier beneath the Rätzligletscher on the northern side. In the first hours of the drainage, the water escaped abruptly, since the moulin reached the bottom of the lake. This drainage phase corresponds to the peak in the discharge curve (≈25 m$^3$ s$^{-1}$) measured in the Simme Valley (blue curve in Fig. 2a). This peak discharge overwhelmed the capacities of the subglacial drainage system, which is indicated by the local ice uplift

measured at all three GPS stations (Fig. 2b). As the lake level fell to the elevation of the moulin inlet, the direct connection between moulin and lake became disrupted. Subsequently, the lake connected to the moulin through a supraglacial channel which steadily incised deeper in to the ice but slowed down the drainage (6-11 m$^3$s$^{-1}$). The exact transition time to this state is unknown but was within the first day of the drainage initiation.

   Discharge magnitudes similar to those of the lake drainage period were also measured in the Simme river prior to the lake

drainage (three peaks on days 213-225) and after the lake drainage (days 248-252). Most of these discharge peaks can be linked to rainfall events having a shorter duration than the lake drainage (precipitation data is provided by the Switzerland's Federal Office of Meteorology and Climatology MeteoSwiss). Since precipitation affects the entire catchment above the gauging station (more than four times the glacier surface area), these precipitation-related discharge events need to be interpreted with caution because part of the measured discharge at those times may be due to water flowing outside of the glacier. In general, however,

the similarity of the discharge curve and the stream level height measured close to the glacier terminus suggests that Glacier de la Plaine Morte is the main contributor to the discharge measurements. In addition to the drainage of Lac des Faverges, a smaller supraglacial lake at the southern rim of the glacier (labeled SL in Fig. 1b) was observed to drain via a supraglacial canyon routing water to moulins. A field visit on day 239 revealed that the lake was draining, but the time of the drainage initiation was not witnessed.





## 3.2 Seismic tremors

Figure 2c shows a spectrogram for station PM05. Recent studies suggest that water routing in subglacial conduits generates seismic tremors observable in the frequency range 1-10 Hz (Bartholomaus et al., 2015; Gimbert et al., 2016). In this frequency

range, however, we observe several signals of anthropogenic origin. These include a diurnal signal from Monday to Friday with sharp onset and decay times, a monochromatic signal visible as spectral line at roughly 2 Hz starting from day 156, and most likely also the diffuse band centered around 5 Hz (Figs. 2c-d). Regarding glacier seismicity, we identify a harmonic moulin tremor with three prominent frequencies which indicate resonant modes in the water column, similar to those in Roeoesli et al. (2016) (Fig. 2d). During the lake drainage, the signal strength is increased for frequencies greater than 1 Hz, and we observe

high-frequency tremors (>3 Hz) during the drainage initiation (Figs. 2c and e).

To better distinguish the seismic signal contributions, we investigate the wavefield in more detail. For this purpose, we calculate 3D particle motion polarization attributes following Koper and Hawley (2010). This approach is based on an eigen-decomposition of the spectral covariance matrix containing the power- and cross-spectra of a single three-component station (Vidale, 1986). One of the polarization attributes, the difference in phase between the vertical and the principal horizon-

tal component, $\phi_{VH}$, allows us to distinguish between different wave types. In particular, the elliptical particle motion of a Rayleigh wave is caused by a 90° phase shift between vertical and horizontal ground motion and distinguishes it from other wave types. To calculate the polarization attributes, we use the freely available toolbox hosted on the IRIS webpage (http://ds.iris.edu/ds/products/noise-toolkit/) with the default parameters and processing steps (including instrument response removal, see Koper and Hawley, 2010). Figure 3 shows probability density functions of $\phi_{VH}$ for a station of each array. Con-

sistent with an elliptical particle motion in the vertical-radial plane associated with Rayleigh wave propagation, $\phi_{VH}$ clusters around ±90° in the frequency range 8.5-12 Hz for all four stations shown. Below 8.5 Hz (6 Hz for station PM33), i.e. frequencies where anthropogenic noise is evident, clustering around ±90° indicative for Rayleigh waves is not present or only in narrow frequency bands (e.g. 4-5 Hz for station PM05). We do not find a difference in the polarization results prior and during the the lake drainage, though the short duration of the drainage process hinders a detailed comparison by means of a statistical

representation as shown in Fig. 3.

Since the continuous recordings below ≈8.5 Hz are contaminated by anthropogenic noise with complex wave-type signature, we chose to analyze the frequency range 8.5-12 Hz in the context of glacial hydraulics. This frequency range is dominated by Rayleigh wave energy which facilitates the tremor location analysis in the next section. To investigate possible correlations between discharge and seismicity for the 8.5-12 Hz range, we calculate the tremor amplitude, or median absolute ground

velocity, for the vertical component of ground velocity as described in Bartholomaus et al. (2015). Figure 4 shows the resulting time series for a station of each array along with the discharge recordings. Prior to the lake drainage, variations in tremor amplitude are weak but follow the discharge curve (e.g. days 213-225). In the four days preceding the lake drainage, the daily melt cycle due to high temperatures is visible both in the discharge time series and the tremor amplitude curves. During and after the lake drainage, tremor amplitudes are increased and show stronger variations than prior to the lake drainage. From Fig. 4(b) we note that PM21's tremor amplitude correlates with discharge particularly well.



## 3.3 Icequake activity

To investigate the interplay of glacial hydraulics and icequake activity on Glacier de la Plaine Morte, we build on the results
of Lindner et al. (2018). This study focuses on ice-fracturing events (Walter et al., 2009) to investigate azimuthal anisotropy

of seismic wave propagation, but the events are also useful to study fracturing associated with glacial hydraulics, e.g. during
outburst floods (e.g. Roux et al., 2010). Lindner et al. (2018) detect icequakes by applying a short-term average/long-term
average (STA/LTA) trigger (Allen, 1978) on bandpass filtered data (10-20 Hz for A1, A2, A3; 7-15 Hz for A0), require at
least three stations of an array to trigger concurrently, and disable the trigger for three seconds after a detected event to avoid
overlapping event windows (for parameter details see Lindner et al., 2018).

From the event catalogs from each array, we calculate the icequake detection rate in events per hour. Figure 5 shows that
icequake activity is often increased during discharge peaks, though not always. Given the correlation between tremor amplitude
and discharge (Fig. 4), this implies that the tremor amplitude in turn is also correlated with the icequake rate (see blue arrows
in Fig. 5). In addition, we identify times where correlation of the tremor amplitude with the icequake rate is higher than with
discharge (red arrows in Fig. 5). These features correspond to icequake bursts lasting on the order of hours but less than a

day. Maximum detection rates are 352, 314, 172, 20 icequakes per hour for A0, A1, A2, and A3, respectively. For A0, this
corresponds to 5.87 icequakes per minute and thus almost 18 seconds of disabled trigger per minute (trigger disabled for 3 s
after event). This suggests that our results are a conservative estimate of icequake occurrence. We note that especially those
arrays with high icqeuake rates (A0 and A1) suffer from icequake contaminated tremor amplitudes. This contamination might
be reduced by choosing different window lengths for the calculation of the tremor amplitude, but investigating this matter is

beyond the scope of this study.

For further insights into the glacial hydraulics, we consider the icequake source locations as determined from plane-wave
beamforming (for details see Lindner et al., 2018). Plane-wave beamforming allows to measure the back azimuth of an incident
signal with an array of sensors. Its generalization to epicentral coordinates will be introduced in Sect. 4.

Figure 6b shows the icequake detections of A1 as a function of focal time and source back azimuth. Some peaks in discharge,

e.g. on day 237, are accompanied by fracturing at various source back azimuths. Since A1 is in the glacier's center and
icequakes arrive from various back azimuths, this suggests that these discharge events affect large portions of the glacier. Other
events, e.g. the melt cycle in the days prior to the lake drainage, are accompanied by more localized seismicity at back azimuths
of 50-100° only. The latter is also the case for the ≈24 hours preceding the drainage initiation, where the seismicity at the back
azimuth towards the main drainage moulin is increased. We detect icequakes at this back azimuth also earlier, but activity is

not sustained and back azimuths do not focus on the moulin. With the onset of the lake drainage, fracturing occurs at various
back azimuths with a focus on the lake basin. After the drainage, fracturing is predominantly confined to the lake basin as well.





## 4 Tremor locations

### 4.1 Matched-field processing

To locate the sustained tremor sources, we apply matched-field processing (MFP, Baggeroer et al., 1993) to our four seismic arrays. MFP measures signal coherence of a phase across an array of receivers and matches it against a synthetic wavefield computed for a point source and a velocity model. By testing various source positions and velocity models for the synthetic

field, a grid search finds the combination of source position and velocity model which best matches the measured coherence across the array. The result is the most likely source location and velocity model. Allowing near-field point sources, hence circular wave fronts, MFP is a generalization of the conventional plane-wave beamforming approach used to determine the slowness and back azimuth of incoming waves (Rost and Thomas, 2002). In case the distance of a source to the array is greater than two to three times the array aperture, the circular wave front approach converges towards a plane-wave solution

(Almendros et al., 1999). For MFP, this implies that far-field sources allow a back azimuth estimate only (as is the case for plane-wave beamforming), while near-field sources can be associated with epicentral coordinates. We leverage this to locate tremor sources some of which are in the the arrays' near-field as we show in the following.

The workflow for MFP is as follows (for a more detailed introduction, see e.g. Corciulo et al., 2012). From the time-domain ground-velocity recordings of $N$ receivers grouped to an array, the discrete Fourier transforms at some angular frequency of

interest $\omega$ are calculated. The resulting complex frequency-domain values are arranged to form a column vector $\mathbf{d}(\omega)$ of length $N$. From this column vector, the cross-spectral density matrix $\mathbf{K}(\omega)$ is calculated as

$$\mathbf{K}(\omega) = \mathbf{d}(\omega)\mathbf{d}^{\dagger}(\omega), \tag{1}$$

where $\dagger$ denotes the complex conjugate transpose operation. Note that the diagonal elements of $\mathbf{K}(\omega)$ are the auto-correlation values of the $N$ receivers at $\omega$, while the off-diagonal elements are cross-correlation values of receiver pairs. Both auto- and

cross-correlations are discrete values associated with angular frequency $\omega$ and the latter represent average phase delays between two receivers at $\omega$. The synthetic field at $\omega$ is calculated for each of the $j = 1, \ldots, N$ receivers as

$$\tilde{d}_j(\omega) = \exp\left(i\omega r_j/c\right), \tag{2}$$

where $i$ is the imaginary unit, $r_j$ is the source-receiver distance of the $j$-th receiver, and $c$ is the phase velocity of the velocity model, which is constant in the case of a homogeneous ice body. Note that this representation focuses on phase information and

disregards amplitude information. For $j = 1, \ldots, N$, the complex $\tilde{d}_j$-values are arranged to form the synthetic column vector $\tilde{\mathbf{d}}(\omega)$ (equivalent to the data vector $\mathbf{d}(\omega)$), and phase matching is achieved via the conventional Bartlett processor (Baggeroer et al., 1993)

$$B_{Bartlett}(\omega) = |\, \tilde{\mathbf{d}}^{\dagger}(\omega)\mathbf{K}(\omega)\tilde{\mathbf{d}}(\omega) \,|, \tag{3}$$



or via a high-resolution MFP method, i.e. the minimum variance distortionless response (MVDR) beamformer (Capon, 1969; Corciulo et al., 2012)

$$B_{MVDR}(\omega) = | \frac{1}{| \tilde{\mathbf{d}}^{\dagger}(\omega)\mathbf{K}^{-1}(\omega)\tilde{\mathbf{d}}(\omega) |} |, \tag{4}$$

where $\mathbf{K}^{-1}$ is the inverse of the cross-spectral density matrix $\mathbf{K}$. In case the incoherent noise power is small relative to the power of the signal of interest, the MVDR processor is capable of estimating the source location and the velocity beneath the

array with higher resolution than the Bartlett processor (Capon, 1969). Note that there is a trade-off between high-resolution and robustness, i.e. in contrast to the MVDR processor, the Bartlett processor might still produce meaningful results if the incoherent noise power is increased.

## 4.2    Single-array results

To investigate the spatial variability of tremor sources across Glacier de la Plaine Morte, we apply MFP to all four arrays

individually. For this purpose, we use a sliding window of 15 minutes length (without overlap) over the entire data set to resolve also temporal variations. Each of these 15-minute segments are processed as follows: To suppress incoherent noise, we calculate the ensemble average of $\mathbf{K}(\omega)$ at discrete frequencies, using a 10 s sliding window with 50% overlap (e.g. Corciulo et al., 2012). The overlap criterion yields a total set of 179 windows over which we average. In the frequency range of 8.5-12 Hz, the results from our polarization analysis (Fig. 3) suggests Rayleigh waves whose amplitude is correlated with discharge. For

this reason, we calculate the MFP results in 0.1 Hz steps and average over the frequency range of 8.5-12 Hz. For the velocity $c$ in Eq. (2), we use the local Rayleigh wave velocities which are 1600 ms$^{-1}$, 1800 ms$^{-1}$, 2200 ms$^{-1}$, and 1600 ms$^{-1}$ for arrays A0, A1, A2, and A3, respectively (Lindner et al., 2018). In the spatial domain, we apply a grid search over the entire glacier surface and its surroundings (assuming a horizontal plane) to calculate the $r_j$ values in Eq. (2) with a spacing of 25 m in x- and y-direction. Figure 7(a) shows the spatial clustering of the MFP results from all available time windows using the MVDR

processor. The picked and shown epicenters are associated with the maximum $B_{MVDR}$ value of all tested coordinates.

### 4.2.1    Array A0

For array A0, three dominant clusters are discernible. Prior to the lake drainage, tremors locate to the north close to the array (Fig. 7, labeled as A0-1) with a few exceptions at high discharge where tremors approach the array from the west. At the drainage initiation, no clear source region can be identified, but with the onset of the drainage, the source locations cluster near

the main drainage moulin (A0-2). This signal remains stable for almost four days before switching again to the source in the north until the end of the drainage. After the lake drainage, the MFP locations cluster predominantly around another moulin in the lake basin which was identified from a high-resolution (0.25 m pixel size) orthophotograph taken on September 7 (by swisstopo, SWISSIMAGE) just after the drainage (A0-3). All three source clusters are located within twice the array aperture and two of them coincide with moulin locations. For this reason, we argue that the MFP locations are robust, event though

uncertainties in epicentral distances increase with distance to the array Walter et al. (2015).





### 4.2.2 Array A1

Prior to the lake drainage, MFP applied to A1 reveals source clustering towards A2 and the small glacier tongue (A1-1). However, on day 232 concurrent with a small peak in discharge and in the five days prior to the lake drainage, another source southwest of the array becomes active (A1-2). In both cases, epicentral distances are not well resolved, which becomes apparent in the elongated clustering in Fig. 7(a) and in the short-term fluctuations of the epicentral distance in Fig. 7(b). In addition, many

source location estimates are beyond the twice-aperture distance, meaning that the curved-wavefront used in MFP converges towards a plane wave which allows back-azimuth estimates of the incoming waves, only (Almendros et al., 1999). During and after the lake drainage, A1 receives signals from two distant sources acting concurrently. One is similar to the dominant source prior to the lake drainage (back azimuth $\approx 320°$) but appears to be associated with slightly increased back-azimuth (5-10°) and distance estimates. The other source originates in the lake-basin direction (A1-3), however, an association with one of the

moulins in this region is due to their similar back azimuths not possible.

### 4.2.3 Array A2

A2 shows less variation than A1 and the source clustering suggests a close tremor source to the northwest of the array in the direction of the small glacier tongue (A2-1). During and after the lake drainage and similar to A1, this source seems to wander slightly farther away towards the north (back azimuth increase of 5-10°). Again, however, epicentral distance is not

well resolved. In some instances and mainly concurrent with discharge peaks, additional signals arrive from a more distant source west of the array (A2-2).

### 4.2.4 Array A3

Tremor signals observed at A3 mainly arrive at the array from the west with back azimuths in the range of approximately 250-285° (A3-1). The epicentral distances of these sources cannot be constrained but their back azimuth points towards a region of

the glacier where several moulins are located. Some of them have a sinkhole-like structure with tens of meters in diameter and are stationary over decades (Huss et al., 2013; GLAMOS, 2018). We also note that a military radar facility is located at a back azimuth of approximately 275° and a distance of around 2 km from the array center, whose operation cannot be excluded as a noise source for A3. Another tremor source is located southwest at a back azimuth of around 230° (A3-2). This source clusters closer than twice the array aperture, is collocated with the position of a moulin identified from ortophotographs, and appears

to be active during peak discharges.

### 4.2.5 Discussion

We also test the MVDR results for plausibility by comparing them to the solutions obtained by using the Bartlett processor. Even though these results were obtained for a smaller spatial grid in order to save computation time, both processors yield similar results. In addition, we also test the robustness of our results by (apart from testing a grid of coordinates) allowing

also a grid search over phase velocity from 1500 ms$^{-1}$ to 3500 ms$^{-1}$ in 50 ms$^{-1}$ steps. Compared to the MVDR-Rayleigh





results, both the back azimuth and the epicentral distance scatter more broadly but the general source distribution stays similar. The velocities for which the Bartlett results are maximized are systematically higher than the Rayleigh wave velocities used previously, especially for A2 (median of approximately 2800 ms$^{-1}$ versus 2200 ms$^{-1}$ for Rayleigh waves). However, the average Bartlett maximum is increased only marginally (0.86 for Bartlett-Rayleigh MFP versus 0.88 for Bartlett MFP with velocity grid search), which indicates that there is a tradeoff between epicentral distance and velocity. Here, we note that

Walter et al. (2015) find quickly growing uncertainties in epicentral distance estimates of icequakes with distance from the array center. The polarization analysis (Fig. 3) suggests that Rayleigh waves are the dominant wave type, though we cannot exclude spurious body wave contributions. Such a contribution could increase the measured apparent velocity due to the higher subsurface velocities of p-waves compared to Rayleigh waves. S-wave velocities in the ice and bedrock (Lindner et al., 2018) are too low to explain the measured velocities at A2. The fact that the median velocities are consistently closer to the expected

Rayleigh wave velocity than to the p-wave velocity in ice ($>3600$ ms$^{-1}$, Podolskiy and Walter, 2016), confirms the polarization results, i.e that Rayleigh wave propagation is dominant.

### 4.3  Multi-array results

To further constrain the tremor source locations, we stack the results obtained from the different arrays. Following the argumentation in the previous section, we continue to focus on Rayleigh waves and consider the MFP results obtained for the

MVDR processor on the entire spatial domain used for the grid search. Figure 8(left column) shows the results for a 15 minute window on day 214 during a peak in discharge caused by precipitation. As reported in the previous section, A0 sees a persistent source to the north of the array, A1 and A2 point towards the glacier tongue, and A3 points toward the (south)west. For A3, however, a secondary lobe of the MVDR output is visible, which points to the glacier tongue as well. We combine the information from different arrays by stacking the MVDR-grids, which results in high MVDR values in regions where multiple arrays

locate signals. The stacking allows triangulation and confirms that the main tremor source is in the region of the glacier tongue (Fig. 8)(left column). We tested other time windows and found that the depicted situation is representative for the pre-drainage period which appears stable with little excursions to other source regions (see Fig. 7)(b). With the onset of the drainage, the tremor source locations change, as shown in Fig. 8(right column). The depicted situation shows the result for a 15-minute window on day 243 roughly 55 hours after the drainage initiation. A0 now locates the tremor signal south of the array and A1

points towards southeast with a secondary lobe pointing to the glacier tongue. A2 again points towards the glacier tongue with less scatter compared to Fig. 8. As discussed in the context of Fig. 7(b), A1 (secondary lobe) and A2 back azimuths are slightly increased compared to the pre-drainage period. Stacking the results from A0, A1, and A2 again (A3 has no data) shows two source regions, the glacier tongue and the main drainage moulin.

### 5  Discussion

To facilitate the interpretation and discussion of the recorded tremors in the context of glacier hydraulics, we first consider the theoretical geometry of subglacial drainage. Figure 9 shows the likely flow paths of subglacial drainage calculated from





the hydraulic potential (Shreve, 1972) for two scenarios: i) englacial water pressures are equal to half of the ice overburden pressure and ii) englacial water pressures are equal to the ice overburden pressure (flotation). Details on the calculation of the hydraulic potential and the shown upstream area distributions which indicate likely subglacial flow paths can be found in Appendix A. Consistent with field observations, both results suggest that almost all water drains through a main outlet beneath Rätzligletscher to the north. At flotation, a second outlet in a few hundred meters to the west of the glacier tongue is visible.

In both cases, the roots of the dendritic network associated with the main outlet are located in both the eastern and western portions of the glacier.

## 5.1    Tremor composition

The results from our tremor analysis demonstrate that the recorded seismic wavefield on time scales beyond those of discrete single events is generated by various processes. Apart from cryo-seismicity, we observe signals of anthropogenic origin. The

diurnal signal occurring on working days only (Fig. 2c; also reported in Preiswerk and Walter (2018)), originates to the south of Glacier de la Plaine Morte (determined by plane-wave beamforming), likely in the Rhone valley where industry is located. The frequency range of anthropogenic noise (e.g. Anthony et al., 2015) often overlaps with the discharge-tremor band, meaning that glacio-seismological data need to be analyzed carefully in glaciated regions with anthropogenic activity such as the European Alps to avoid misinterpretation. This also holds in the absence of anthropogenic noise, since our data reveal that

tremors may be generated by different aspects of glacier hydraulics at the same time. We identify tremors which are dominated by energy released through ice fracturing (A0 and A1), locate at moulin locations (A0 and A3), or exhibit a characteristic frequency signature of moulin resonances (A3) and thus obscure turbulent-flow tremors. However, at A2, we argue that the recorded tremors are generated by subglacial water routing for the following reasons: i) the tremor amplitude correlates with the discharge curve (Fig 4) and ii) MFP shows a persistent source in the region of the glacier tongue (Fig. 7a), from where iii)

the main glacier outlet emerges (Fig. 9). We note that subglacial water routing in turn can generate tremors both via pressure fluctuations in turbulent flow and via impact events from bedload sediment transport (Gimbert et al., 2014, 2016). Recent studies (Bartholomaus et al., 2015; Gimbert et al., 2016) typically separate the two processes by frequency at around 10 Hz, thus the frequency range associated with our results (8.5-12 Hz) may contain both processes. Even though we cannot exclude that bedload significantly contributes to total seismic power, we see evidence for water tremors being the dominant source for

the following reasons. i) The frequency ranges are controlled by various parameters (channel to station distance and channel apparent roughness among others) permitting also turbulent-flow tremors above 10 Hz (Gimbert et al., 2014). ii) Ice flow of Glacier de la Plaine Morte is negligible ($< 1$ cm/day at A2; not shown), resulting in little sediment production by abrasion (Hallet, 1979), which we expect hinders bedload tremor generation. iii) The A2 tremor-discharge scaling as discussed later tends to follow the drainage-regime predictions for water tremors without evidence for a hysteresis due to sediment flushing

(e.g. Gimbert et al., 2016). Apart from various tremor sources, we finally note that the tremors are composed of different wave types, further increasing the complexity of the tremor signal.





## 5.2 Temporal evolution of the drainage system

Figure 7(b) shows that the tremor locations change over time. Since icequake tremors typically last on the order of hours (Fig. 5) but the inferred back azimuths of sources stay stable on the order of days to weeks, we attribute the dominant source locations to moulin tremors and subglacial water routing. At the end of July (when the deployment of all sensors was completed), A1 and A2 tremor sources locate towards the glacier tongue, and A3 tremor sources in the region to the west of the array where

multiple moulins are located. In addition, we note that seismic tremors are likely generated by efficient channelized subglacial flow (Gimbert et al., 2016) and that the moulins in the vicinity of A3 seemed to evacuate meltwater without the build-up of supraglacial lakes or reservoirs. We therefore suggest that the left branch of the upstream area distributions in Fig. 9, or more general the western and northern part of the glacier, had an efficient and channelized subglacial drainage system. According to Fig. 7(b), this configuration stayed stable until end of August (day 236). At the same time, A0 saw a persistent source to the

north whose origin remains elusive. A potential explanation for this source could be another moulin feeding the upstream area branch (Fig. 9) originating in the northeast of the glacier. However, neither the observations from our regular station visits, nor the orthophotograph show evidence for a moulin in the area of A0. Starting on day 236, A1 points toward the southwest (Fig. 7) and we attribute this source to the drainage of the smaller supraglacial lake (SL in Fig. 1b). With the onset of the drainage of Lac des Faverges, tremors from the lake basin become dominant (A0 and A1), suggesting that the eastern part of the glacier

has an efficient connection to the drainage system, as tremors are expected to originate from channelized flow (Gimbert et al., 2016). Combining this information with the theoretical pattern of subglacial water routing (Fig. 9) suggests that the seismic tremors reveal a gradual "upglacier" (along the main branch in Fig. 9 from north to south to east) evolution of an efficient channelized drainage system as the melt season progresses. This matches both the field observations (first SL connects to the drainage system, then Lac des Faverges) and the theory of subglacial channel evolution throughout a melt season (Werder et al.,

20  2013).

While subglacial channel evolution is typically described through the competing mechanisms of melting and ice creep (Röthlisberger, 1972), our results show that fracturing can play an important role under specific flow scenarios. We find that icequake activity in the lake basin precedes the drainage onset by several hours (Fig. 6). In combination with a lake reservoir which pressurizes the void spaces and the englacial environment, we suggest that hydrofracturing (e.g. Van Der Veen, 1998;

Roberts et al., 2000) drives the drainage initiation. Since no sustained seismicity in the lake region is detected prior to that, this highlights the potential of passive seismic monitoring for early warning of glacier-dammed lake outburst floods. Apart from the lake drainage, other discharge peaks are accompanied by fracturing as well (Figs. 5 and 6). However, we note that elevated strain rate resulting from water-enhanced basal sliding may give rise to icequakes as well (Podolskiy et al., 2016).

## 5.3 Drainage regime

### 5.3.1 Theory

Water flow through ice-walled conduits is driven by the hydraulic pressure gradients along the conduits. Along the channel walls, frictional heat enlarges the channels. At the same time, ice creep closes the conduits in case the ice overburden pressure





exceeds the water pressure in the conduit (Röthlisberger, 1972). These two counteracting processes result in a temporal evolution of conduit radius and water pressure in the conduit. Recently, Gimbert et al. (2016) suggested that pressure fluctuations due to turbulent flow in subglacial conduits can generate seismic tremors whose power scales with discharge according to the drainage regime. Gimbert et al. (2016) derive two end member scenarios for which the relative seismic power $P_{rel}$ and relative discharge $Q_{rel}$ (with respect to some reference state) are related through a power law but with different scaling exponent.

5    (i) Varying hydraulic pressure gradient and constant hydraulic radius, implying $P_{rel} \propto Q_{rel}^{14/3}$. As defined in Gimbert et al.
        (2016), changes in the hydraulic pressure gradient are caused by variations in the water pressure $p$ along a conduit, i.e.
        $\partial p/\partial x$, where x is the distance along the channel. Such a situation is schematically depicted in Fig. 10, where e.g. the
        diurnal melt cycle causes hydraulic head variations in a moulin without changes in the hydraulic radius of the conduit.
        At some distance from the moulin, at the glacier snout, water constantly flows at atmospheric pressure. As the hydraulic
10      head in the moulin varies, this results in pressure gradient changes in the subglacial conduit implying $P_{rel} \propto Q_{rel}^{14/3}$.
        This drainage regime is expected to dominate in filled subglacial conduits which do not adjust their hydraulic radii fast
        enough to accommodate discharge changes. We expect that this occurs e.g. for strong daily melt variations in the early
        melt season (when the capacity of the conduits is still limited) or for rapid water input due to a sudden lake drainage.

    (ii) Varying hydraulic radius and constant hydraulic pressure gradient, implying $P_{rel} \propto Q_{rel}^{5/4}$. As the hydraulic radius of a
15       conduit is defined as its cross-sectional area divided by the wetted perimeter, both changes in the water level of a conduit
         operating under atmospheric pressure and the cross-section of a fully filled conduit result in variations in the hydraulic
         radius (Fig. 10). For instance, subglacial water routing at atmospheric pressure is predicted to be revealed by the power
         law $P_{rel} \propto Q_{rel}^{5/4}$ as the wetted perimeter can vary without geometrical changes in the subglacial conduits. The same
         scaling relationship holds for filled conduits, in case melt enlargement and creep closure of channels dominate over
20       changes in the pressure gradient.

Gimbert et al. (2016) also derived solutions for the relative hydraulic pressure gradient $S_{rel}$ and the relative hydraulic radius $R_{rel}$ (again with respect to some reference state) as a function of observed $P_{rel}$ and $Q_{rel}$, given as (Gimbert et al., 2016, eq. 5)

$$S_{rel} = P_{rel}^{(24/41)} Q_{rel}^{(-30/41)} \tag{5}$$

$$R_{rel} = P_{rel}^{(-9/82)} Q_{rel}^{(21/41)}. \tag{6}$$

### 25 5.3.2 Observations

At A2, we observe tremors due to subglacial water flow beneath Rätzligletscher. Knowing the source locations of subglacial tremors allows us to apply the tremor-discharge relationships more targeted as without that knowledge. If the source locations are not known, the tremor-discharge scalings provide an integrated view over the surroundings of the seismic measurements, whereas the locations presented in Fig. 7 allow us to investigate glacier hydraulics at a specific point, i.e. beneath
30 Rätzligletscher. As Rätzligletscher accommodates the main outlet, we argue that discharge measured in the Simme valley is





representative for water routing at the measured A2 tremor locations. Furthermore, we expect that the number of conduits close to the outlet stays constant. Both assumptions favor the successful application of the tremor-discharge relationships.

Figure 11 shows the scaled seismic power $P_{rel}$ (square of the tremor amplitude) versus the scaled discharge $Q_{rel}$ (using the minimum discharge value and its associated seismic power for scaling) on a log-log plot (for details see appendix B). In this representation, the slope equals the exponent $x$ of $P_{rel} \propto Q_{rel}^x$, where the black lines indicate discharge routing accommodated

by hydraulic radius adjustment ($x = 5/4$) and the red lines discharge routing accompanied by variations in pressure gradient ($x = 14/3$), respectively.

In the pre-drainage period (Fig. 11(a)), the power-discharge representation shows a general trend towards radius-adjusting conduits. This is also revealed by the $x$-exponent distribution (upper right in Fig. 11(a)) obtained by calculating the slopes between two adjacent samples. This in turn implies that pressure-gradient adjustment occurs seldom and on short time scales

only. Such a system is indicative of a well-established, channelized drainage system evacuating water efficiently without significant pressurization. We find such a configuration on Glacier de la Plaine Morte, where the source region of the tremors corresponds to the main trunk of an arborescent drainage network (indicated by the upstream area distributions). However, for the approximately ten days preceding the drainage but in particular for the last four days of this time span with a pronounced diurnal melt cycle, the data suggest pressure gradient adjustments (yellow dots). This indicates that the capacity of the conduits

cannot yet accommodate the water from the melt events without pressurization.

Fig. 11(b) shows the power-discharge scaling for the drainage period. At the drainage onset, the data points scatter along the pressure-gradient adjustment prediction (black dots). Subsequently, after a more chaotic phase associated with clockwise hysteresis, the data reveal hydraulic-radius adjustments during most of the drainage period (purple and orange dots), which is again followed by pressure-gradient adjustments at the end of the drainage (yellowish dots).

To investigate these observations in more detail, we consider the evolution of the hydraulic pressure gradient and the hydraulic radius as calculated from Eq. (5) and (6), respectively. In Fig. 12, we compare $R_{rel}$ and $S_{rel}$ to the measurements of the lake level and the ice surface uplift, which also provide constraints on the drainage hydraulics. In addition, the pictures of the automatic camera provide an estimate of the time when the lake basin was empty. As already inferred from Fig. 11, the diurnal melt cycles prior to the drainage cause pressure-gradient variations while the hydraulic radius changes little. In this phase, the

daily peaks of the pressure gradient occur around the time of maximum daily discharge. At the onset of the lake drainage in the evening of day 240 as the lake level starts to drop (gray dashed line in Fig. 12), the inferred pressure gradient increases and reaches its maximum when the rate of ice uplift at A2 is highest. At the same time, the hydraulic radius is described by a transient decrease. Subsequently, the pressure gradient decreases to high pre-drainage values. Concurrently, the hydraulic radius increases as the discharge increases. After the peak discharge, the hydraulic radius decreases again but remains above

the pre-drainage level. Subsequent variations in the hydraulic radius and the pressure gradient stay on an elevated level. The sharp peak and drop in $S_{rel}$ and $R_{rel}$ on day 243, respectively, correspond to the time where we re-installed the A2 stations directly on ice, as the snow cover was diminishing. According to the imagery, the emptying of the lake basin was finished in the night from day 246 to 247 (gray dashed line in Fig. 12). A few hours earlier, discharge starts to drop to pre-drainage values.





We observe the same for the hydraulic radius. In contrast, the pressure gradient briefly increases before dropping to values lower then prior to the drainage.

### 5.3.3 Interpretation

From all our measurements, we deduce the following history of glacier hydraulics associated with the drainage. In the hours prior to the drainage onset, the lake reaches the drainage moulin but the latter is not yet connected to the subglacial drainage

system (situation schematically depicted in Fig. 13a). At this stage, seismic tremors are generated beneath the glacier tongue by the 'background' meltwater routing where the daily melt events cause daily variations in the pressure gradient. Through hydrofracturing (section 3.3), the moulin then connects to the subglacial drainage system causing a sudden water input into the drainage system. The lake discharge overwhelms the drainage system, as "an excess of water is pouring into a conduit system of low capacity" Röthlisberger (1972), which results in a pressurization of the subglacial environment. From our GPS

measurements, it is evident that water pressures exceed the ice overburden pressure, which results in local flotation. The pressure gradient, in turn, can be approximated as the difference in pressure on either side of the tremor generating region. Considering that water is at atmospheric pressure at the outlet of the glacier tongue, an increase in subglacial pressure due to the lake drainage would also cause an increase in pressure gradient as illustrated in Fig. 13. This is in agreement with our power-discharge derived pressure gradient history. Since the conduits cannot adjust their size fast enough, discharge increases

only slightly as the lake level starts to drop (Fig. 12). Subsequently, the cross-sectional area (and thus the hydraulic radius) of the subglacial conduits increase due to frictional heat of pressurized flow causing melting of the ice walls (Röthlisberger, 1972). As the conduits increase in size allowing larger discharge, water is effectively evacuated resulting in a drop in the pressure gradient and causes the ice uplift to cease (Fig. 12 and Fig. 13c). The time scale of conduit enlargement due to melting is expected to be on the order of hours to days (Mathews, 1973).

In the following, as the lake steadily spills water into the moulin, the conduits adjust their size by the competing mechanisms of closure due to ice creep (Nye, 1953; Glen, 1955) and opening due to melting, without significant pressurization and radius changes. Finally, as the discharge drops at the end of the lake drainage, the conduits decrease their size. As the conduits close, another short phase of pressure build up occurs, indicating the capacity of the conduits is decreased too quickly to maintain constant pressures (Fig. 13d). We expect that conduits tend to close due to ice creep as discharge decreases at the end of the

drainage. However, we note that the contraction of conduits takes place on the order of days to weeks, especially for thin ice (smaller 100 m) as encountered on Glacier de la Plaine Morte (Mathews, 1973). This suggests that our inferred closure rates of the relative hydraulic radius (Fig. 12) might be overestimated. Another explanation could be the physical collapse of parts of the conduits as discharge decreases (Mathews, 1973). Figs. 5 and 6 show that fracturing is indeed pronounced at the end of the lake drainage but we cannot find evidence for strong fracturing from the direction of the glacier tongue, which is expected

for mechanical failure during conduit collapse. In addition, also the drop in hydraulic radius at the onset of the lake drainage remains enigmatic as we do not have a reason to believe that conduits shrink as an ice-marginal lake starts to drain. We suggest that this drop is an artefact that could be due to neglecting potential changes in channel number and position when inverting for $S_{rel}$ and $R_{rel}$ using Eq. (5) and (6) or by not accounting for sheet-like flow during the ice uplift phase.



## 6  Conclusions

In this study, we analyzed the seismicity on a plateau glacier in the Swiss Alps in the context of glacier hydraulics. We find that the nature of glaciohydraulic tremors is time dependent and shows spatial variability on the sub-kilometer scale. The tremors are generated by subglacial water flow, icequake bursts, or in moulins. By combining our seismic analysis with upstream area distributions of subglacial flow, we find that the tremors indicate the gradual evolution of an arborescent drainage system and

that the lake drainage is initiated by hydrofracturing. The fracturing is a precursor of the drainage and might be used for early warning though we cannot generalize this for all outburst floods. To investigate the drainage regime, we focused on tremors originating beneath the glacier tongue. At the onset of the lake drainage, the tremor-discharge analysis suggests a pressurization of the subglacial environment, which is followed by an enlargement of sugblacial conduits. Measurements of the ice surface motion (through GPS) and the lake level support the drainage-regime history inferred from passive seismic measurements

conducted at the ice surface combined with discharge data. Our source locations allow a spatio-temporal investigation of the subglacial drainage system and highlight the use of cryoseismology with respect to glacier hydraulics.

*Data availability.*  Seismic data used in this study are accessible via the repository of the Swiss Seismological Service under network code 4D. GPS data are available upon request. Discharge data are available via Switzerland's Federal Office for the Environment and precipitation data via the Federal Office of Meteorology and Climatology MeteoSwiss.

## Appendix A:  Subglacial drainage

Beneath glaciers, water flows in response to the hydraulic potential $\phi$, which is the sum of the pressure potential and the elevation potential (Shreve, 1972), i.e.

$$\phi = f\rho_i g h_i + \rho_w g z_b,\tag{A1}$$

where $f$ is the flotation fraction, $\rho_i = 910$ kgm$^{-3}$ and $\rho_w = 1000$ kgm$^{-3}$ are the densities of ice and water, $g = 9.81$ ms$^{-2}$

is the gravitational acceleration, $h_i$ is the (laterally varying) ice thickness and $z_b$ is the bedrock elevation. Measurements of the ice thickness are available along a grid of flight profiles where the glacier bed was surveyed with helicopter-borne ground-penetrating radar (GPR, Langhammer et al., 2018; Grab et al., 2018). We interpolate the ice thickness values available along the GPR-profiles to a regular 50-m-grid using inverse distance weighting (Shepard, 1968) of the 100 nearest data points and their corresponding ice thicknesses. In addition to the GPR profiles, we also use the coordinates of the glacier margin (e.g. Fig. 1) for the interpolation, where we set the ice thickness to zero. We then calculate the bedrock topography by subtracting the ice

thickness from the digital elevation model. Subsequently, we calculate the hydraulic potential for $f = 1.0$ (water pressure equals the ice-overburden pressure), since we expect high water pressures, especially during the lake drainage initiation (Roberts, 2005). This is confirmed by continuous GPS measurements in the vicinity of A0, A2, and A3, which show vertical lifting





during the first ≈8-36 hours of the lake drainage (Fig. 2b). In addition, we also consider the hydraulic potential calculated for (spatially uniform) water pressures of half the ice overburden for comparison.

To investigate likely subglacial water-flow paths, we calculate the upstream area for each grid cell, i.e. the (grid cell) area that is upstream and connected to the grid cell of consideration. We follow the approach of (Flowers and Clarke, 1999) and calculate the upstream area distribution using the Quinn algorithm (Quinn et al., 1991), which transfers the area to all downstream cells
among the eight direct neighbor cells weighted by the relative gradients. We perform depression filling of the hydraulic potential surfaces and subsequent calculation of the upstream area using the RichDEM toolbox (Barnes, 2016). While the results might suffer from inaccuracies introduced by the interpolation of the ice thickness profiles and by neglecting (horizontal) englacial transport as well as subglacial mechanics (Flowers and Clarke, 1999), they are consistent with field observations (see main text for details).

**Appendix B: Tremor-discharge scaling**

Discharge data is provided in hourly averages, while tremor amplitude samples are calculated from 30 minutes of data with 50 percent overlap, resulting in a sample spacing of 15 minutes. For consistency and to smooth the (partly) noisy tremor data (Fig. 4(b)), we also calculate running averages of the tremor amplitude by taking a window of five samples centered around each timestamp associated with the discharge data. In addition, we test corrections of the discharge time series for the time it
takes the water from the glacier terminus to the gauging station (≈4.5 km horizontal distance and ≈1.5 km elevation difference) by up to two hours travel time but this does not change our conclusions.

*Author contributions.* FL, FW, and GL designed the experiments, which were carried out by all authors. FL processed and analyzed the data with the help of FW and FG. FL prepared the manuscript with contributions from all co-authors.

*Competing interests.* The authors declare that they have no conflict of interest.

*Acknowledgements.* This work was funded by the Swiss National Science Foundation project Glacial Hazard Monitoring with Seismology (GlaHMSeis project PP00P2 157551). The geophones and DataCubes for 15 stations were provided by the Geophysical Instrument Pool Potsdam (GIPP) under project AnICEotropy. We thank Andreas Bauder who installed the GPS stations and Philippe Limpach who processed the GPS data, as well as Lorenz Meier from Geopraevent for sharing the data from the lake monitoring. We are also grateful to our technicians Pascal Graf and Christian Scherrer, as well as to Adrian Doran and all other people who helped in the field. We appreciate the collaboration
with the municipality of Lenk and would like to acknowledge logistical support from Remontées Mécaniques de Crans-Montana (CMA), the Swiss Armed Forces, and the Federal Office of Civil Aviation. We used ObsPy (Beyreuther et al., 2010) for seismic data processing and created the figures with the Matplotlib plotting library for Python (Hunter, 2007).



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





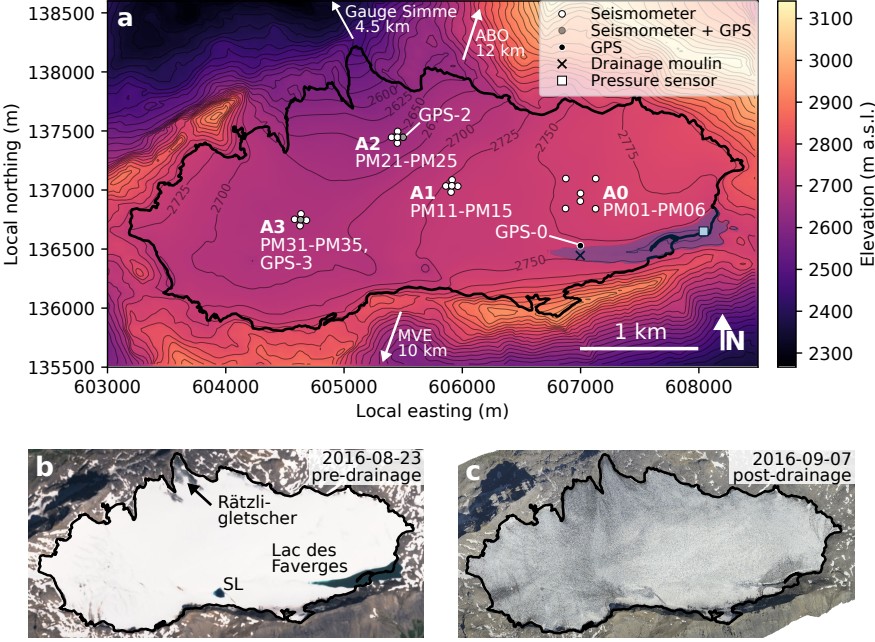

**Figure 1.** (a) Map of the extent of Glacier de la Plaine Morte (thick black line), topography (contour lines and color-coding), and location of sensor installations (symbols). Seismic stations are numbered for each array (A0-A3) counterclockwise from 1 (north; northeast for A0) to 5 (center station). Station PM06 (lower center station of A0) was added end of July. The blue shaded area depicts the approximate maximum extent of Lac des Faverges with the moulin through which the drainage initiated (black cross). The arrows indicate the direction and distance to the discharge gauge of the Simme river and to the weather stations ABO and MVE. (b) Sentinel-2 imagery (modified Copernicus Sentinel data 2019/Sentinel Hub) acquired on 2016-08-23 (day 236) with the glacier extent from (a). SL stands for supraglacial lake. (c) Orthophoto taken on 2016-09-07 (day 251) with the glacier extent from (a).

**Figures**



**Figure 2.** (a) Hydrological data recorded in the vicinity of Glacier de la Plaine Morte: discharge of the Simme river measured in the Simme Valley (blue curve; ≈4 km line of sight from the glacier terminus), level of the Truebbach stream (gray curve; ≈1.5 km from glacier terminus), height of the water column in the lake above the pressure sensor (orange dashed), precipitation at stations ABO and MVE (blue and purple bars; 12 km north and 10 km south of the glacier, respectively). (b) Discharge and lake level for the drainage period (same as (a)) and the vertical displacement of three GPS units (black lines). (c) Spectrogram of station PM05 for the same time period shown in (a). (d) and (f) Zoom-in of (c) showing anthropogenic noise and the lake drainage, respectively. (d) Zoom-in of a spectrogram of station PM32 showing moulin resonances. Note that data used to calculate the spectrograms are not corrected for the instruments' phase responses.

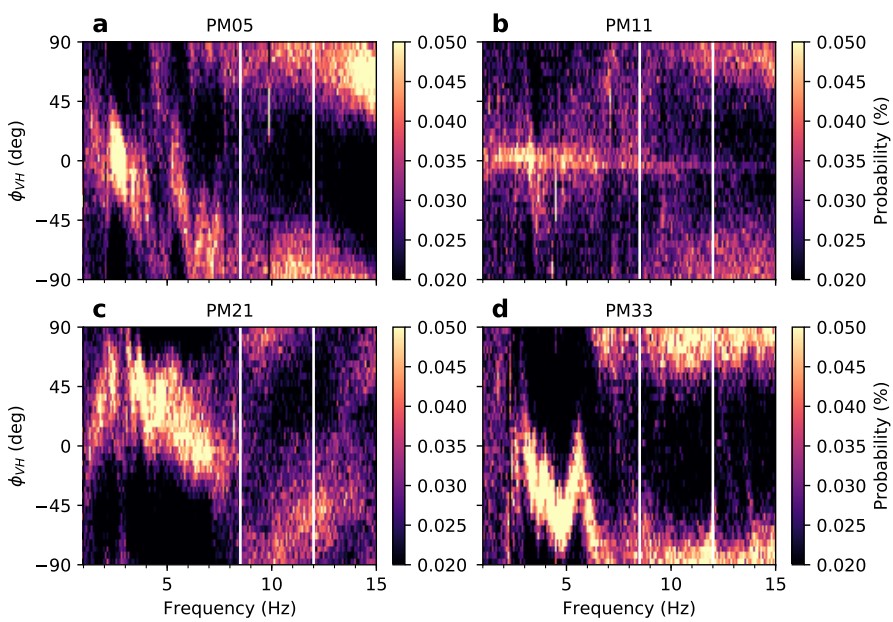

**Figure 3.** Probability density functions for the phase difference between the vertical and principal horizontal component. Probabilities are calculated for the time period 2016-07-22 to 2016-09-06 (2016-08-23 for station PM33) and bins of five degrees width. The results are shown for one station of each array: (a) PM05 , (b) PM11, (c) PM21, and (d) PM33.

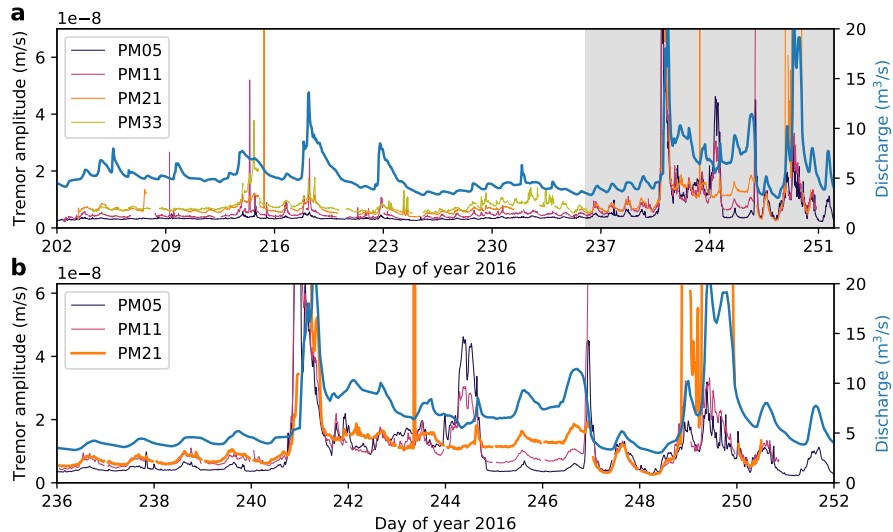

**Figure 4.** (a) Tremor amplitude (8.5-12 Hz) time series for a station of each array (thin colored lines) and discharge (thick blue line). (b) Zoom-in on the gray shaded area in (a).





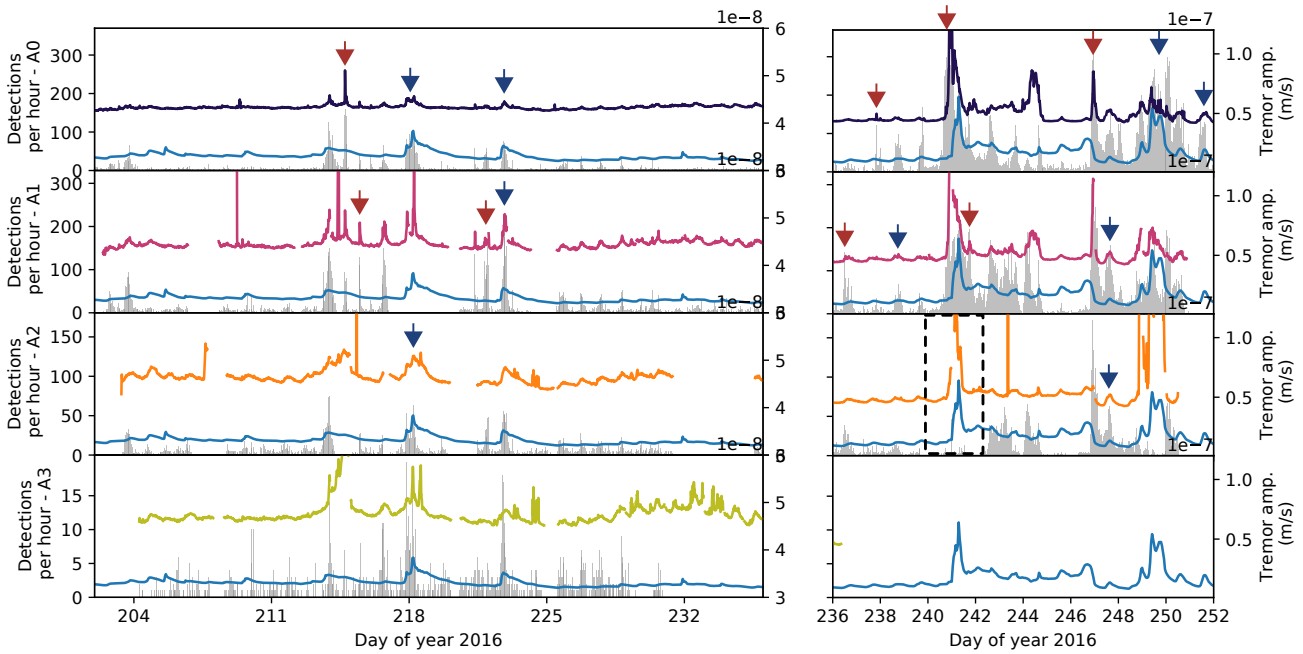

**Figure 5.** Icequake detections per hour for all arrays (gray bars; see text for details) and the discharge curve of the Simme river (blue line). The black, magenta, orange, and green lines (from top to bottom) are the tremor amplitude curves shown in Fig. 4. Note the different tremor amplitude scaling between the two panels. Blue arrows indicate times where the tremor amplitude correlates with both discharge and icequake rate. Red arrows indicate times where the tremor amplitude correlates with icequake rate, only. The black-dashed rectangle indicates times, where three of five A2 stations tipped over due to diminishing snow cover. The icequake rates in this interval need to be taken with caution.

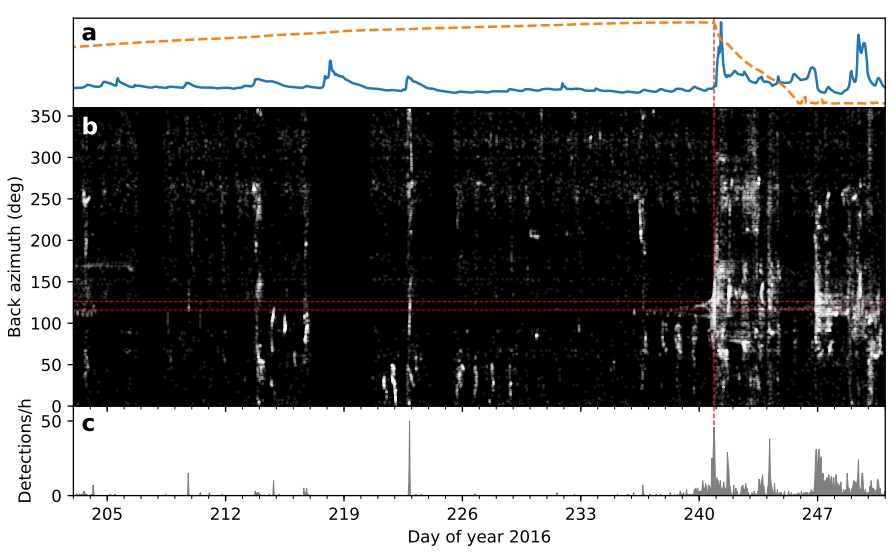

**Figure 6.** (a) Discharge and lake level (same as in Fig. 2). The vertical red-dashed line indicates the drainage initiation. (b) Detected icequakes at A1 as a function of time and source back azimuth (white dots on black background). Icequake clustering both in time and back azimuth is visible as bright white spots. The two horizontal red-dashed lines indicate the the back azimuth from the array center to the main drainage moulin $\pm 5°$. (c) Icequakes per hour in the back azimuth range marked with the two horizontal red-dashed lines in (b).



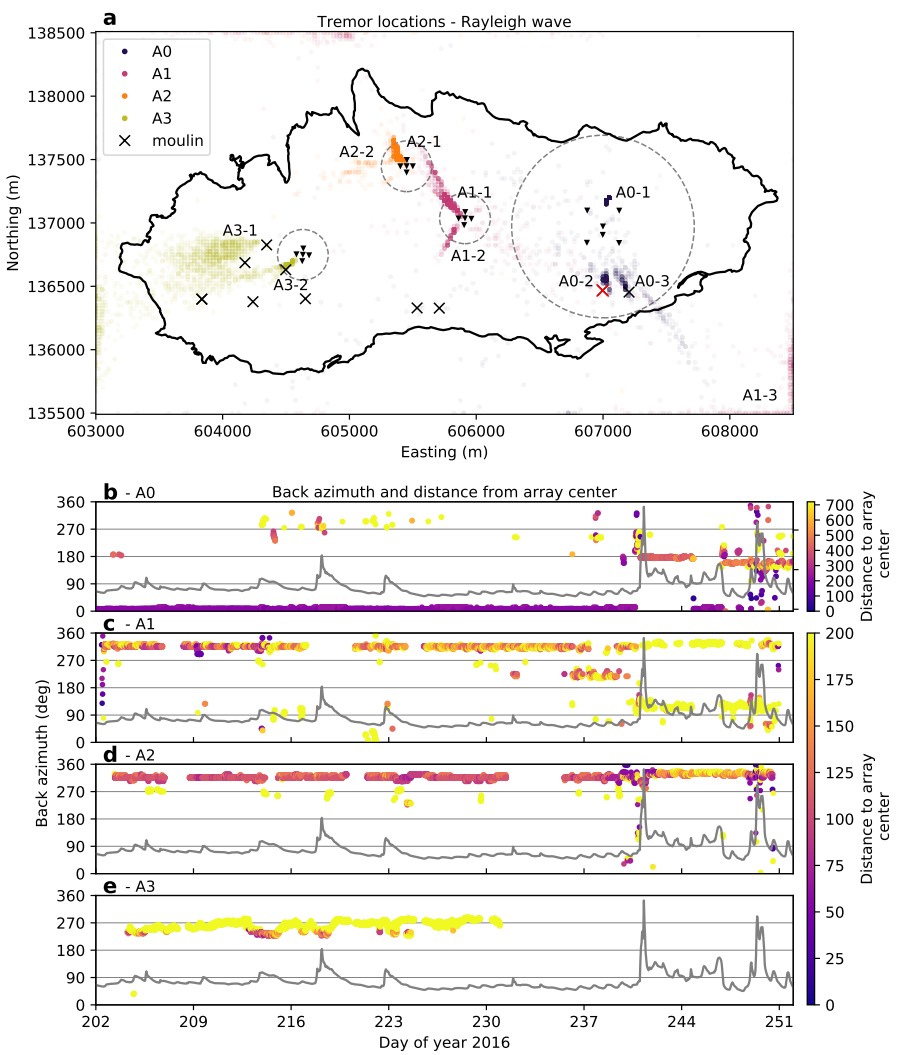

**Figure 7.** (a) MFP locations (MVDR processor) over the frequency range 8.5-12 Hz assuming Rayleigh wave velocities (colored dots). Each dot is the result obtained for a 15-minute window. The thick black line indicates the glacier margin in 2015, the black triangles the locations of the seismic stations, and the gray dashed circles the distance of twice the array aperture from the array center. The black crosses indicate positions of moulins identified from orthophotographs (by swisstopo, SWISSIMAGE). The red cross marks the position of the lake drainage moulin. Labels of type A0-2 refer to dominant source clusters discussed in the text. (b)–(e) Temporal variation of back azimuth and distance of the tremor source locations (colored dots) from (a) as seen from the array centers of A0, A1, A2, and A3, respectively. The grey line depicts the discharge curve measured in the Simme Valley for reference.





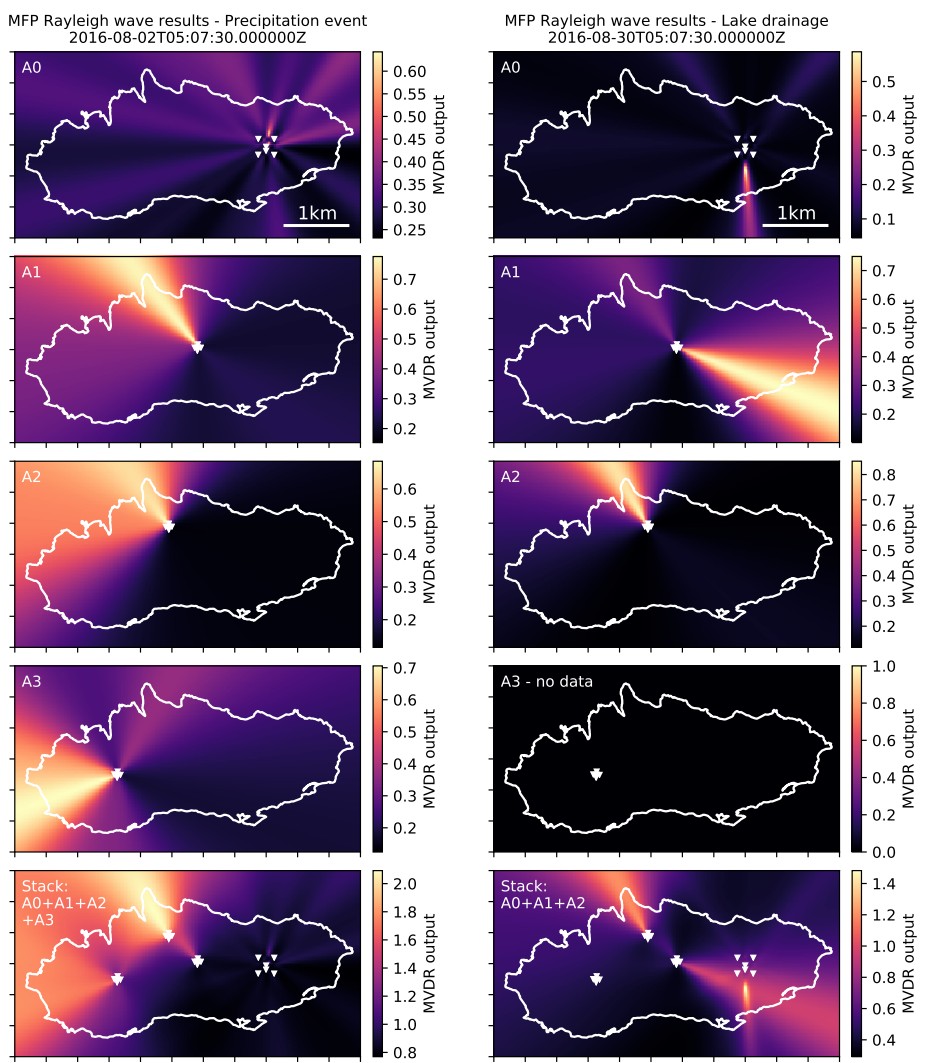

**Figure 8.** MFP results obtained using the MVDR processor and Rayleigh wave velocities. Shown are the results for the single arrays (rows 1-4) and a stack of the arrays (last row). The left column shows the results of a 15 minute window on day 214 during a peak in discharge caused by precipitation. The right column depicts the results of a 15 minute windows during the lake drainage on day 243. Exact times are given on top of the plots. The spacing of the ticks on the x- and y-axis is 500 m (see also Fig. 1).





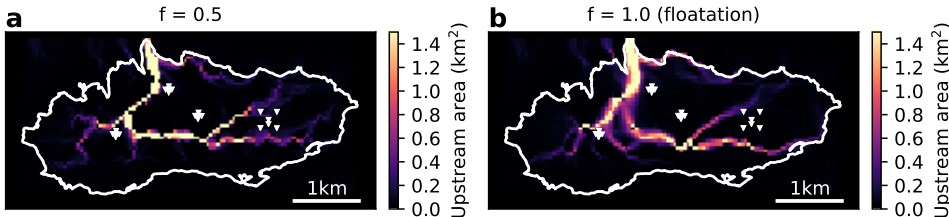

**Figure 9.** Upstream area distributions calculated from the hydraulic potential (see text for details). (a) Solution obtained for (spatially uniform) water pressures of half the ice overburden pressure. (b) Solution obtained for (spatially uniform) water pressures equaling the ice overburden pressure. The whit triangles indicate the positions of the seismic stations.

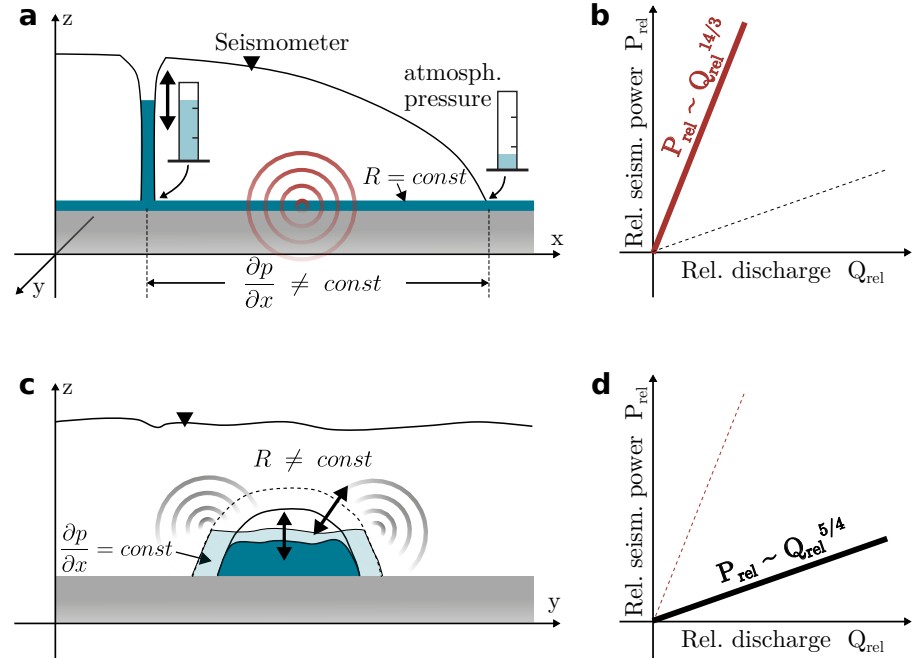

**Figure 10.** Interpretation of the theory of Gimbert et al. (2016) relating seismic power to discharge. (a) Cross-section through a glacier parallel to the flow direction. The hydraulic head in the moulin varies due to e.g. the daily melt cycle. The hydraulic radius of the subglacial conduit is constant. As the water routing at the glacier snout occurs at constant atmospheric conditions, a pressure gradient in the subglacial conduit is present. (b) For such a configuration of varying hydraulic pressure gradient (and constant hydraulic radius) the relative seismic power is predicted to scale with the relative discharge (relative to some reference state) to the power of $14/3$. (c) Cross-section perpendicular to the flow direction. The hydraulic radius of a subglacial conduit varies through a change in water level or through changes in the cross-sectional area due to frictional melting or creep closure. The pressure gradient is assumed constant. (d) For such a configuration of varying hydraulic radius (at constant hydraulic pressure gradient), the relative seismic power is predicted to scale with the discharge to the power of $5/4$.



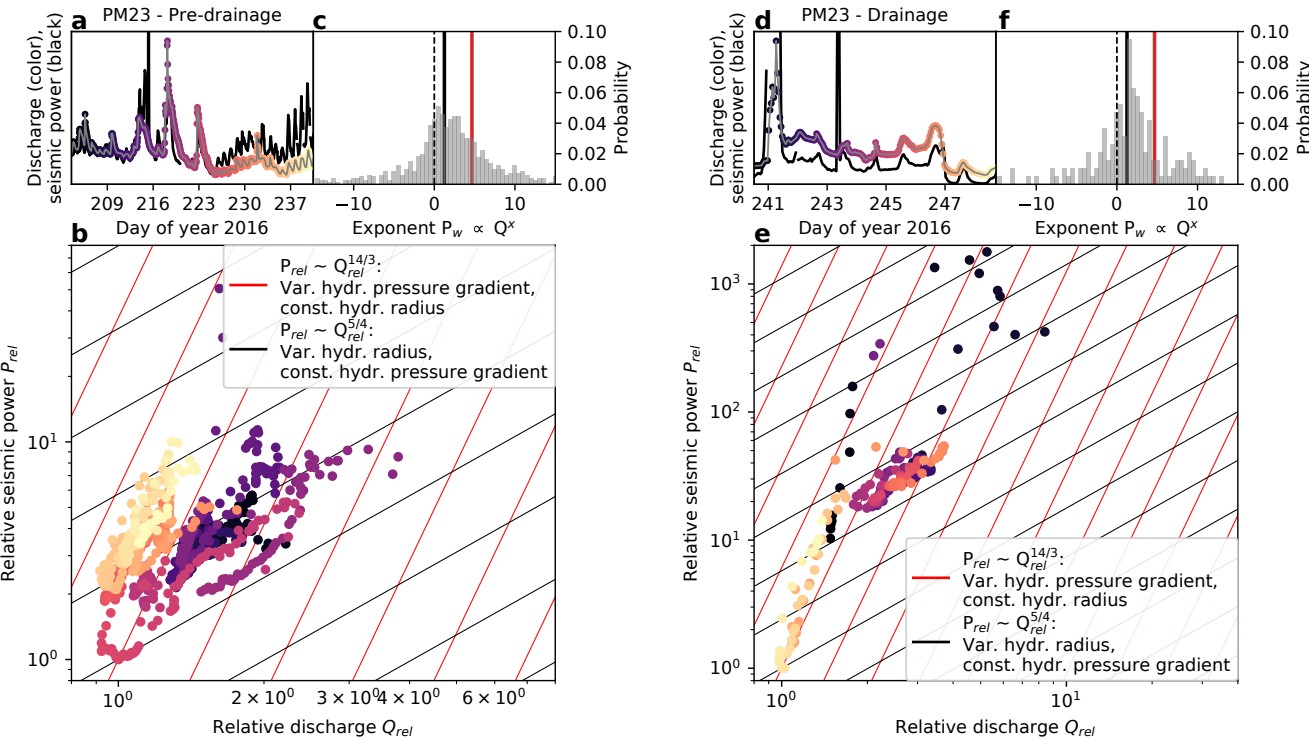

**Figure 11.** Tremor-discharge scaling of station PM23. (a) Tremor amplitude (black) and discharge curve (colored) for the pre-drainage period. (b) Scaled seismic power as a function of the scaled discharge (see text for details) on a log-log plot. Color-coding corresponds to the plot in the top left. Red and black lines are the drainage-regime predictions of Gimbert et al. (2016) and indicate discharge routing through variations in the hydraulic pressure gradient and variations in the hydraulic radius, respectively (see legend). (c) Distribution of slopes (and thus exponents) calculated from the log-log representation of two adjacent samples each. Black and red bars again show the expected exponents for the two drainage regimes (see legend in (b)). (d)–(f) Same as (a)–(c) but for the drainage period.



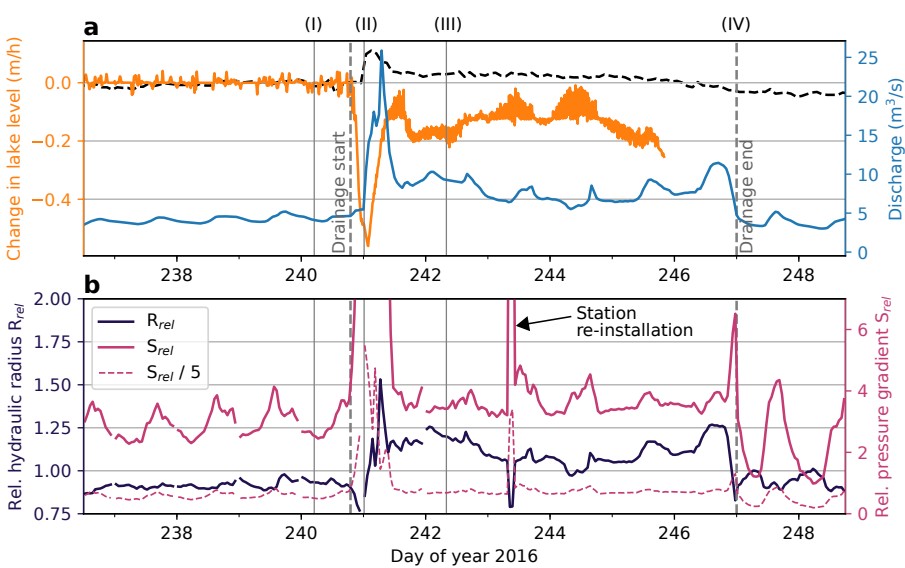

**Figure 12.** (a) Change in lake level (orange), discharge measured in the Simme valley (blue), and vertical ice surface motion at A2 (GPS-2, black dashed). Maximum ice uplift is around 5 cm, see Fig. 2b for scale. The vertical grey dashed lines indicate the start and end time of the drainage, as determined from the lake level change and the automatic camera (as the lake level sensor was not installed at the deepest point of the basin and thus did not provide measurements until the end of the drainage), respectively. Vertical grey bars and roman numbers (I)-(IV) mark snapshots illustrated in the cartoon in Fig. 13. (b) Evolution of the relative hydraulic radius (black) and the relative pressure gradient (magenta) derived from the seismic power and the discharge curve (eq. 6) for the same time period.



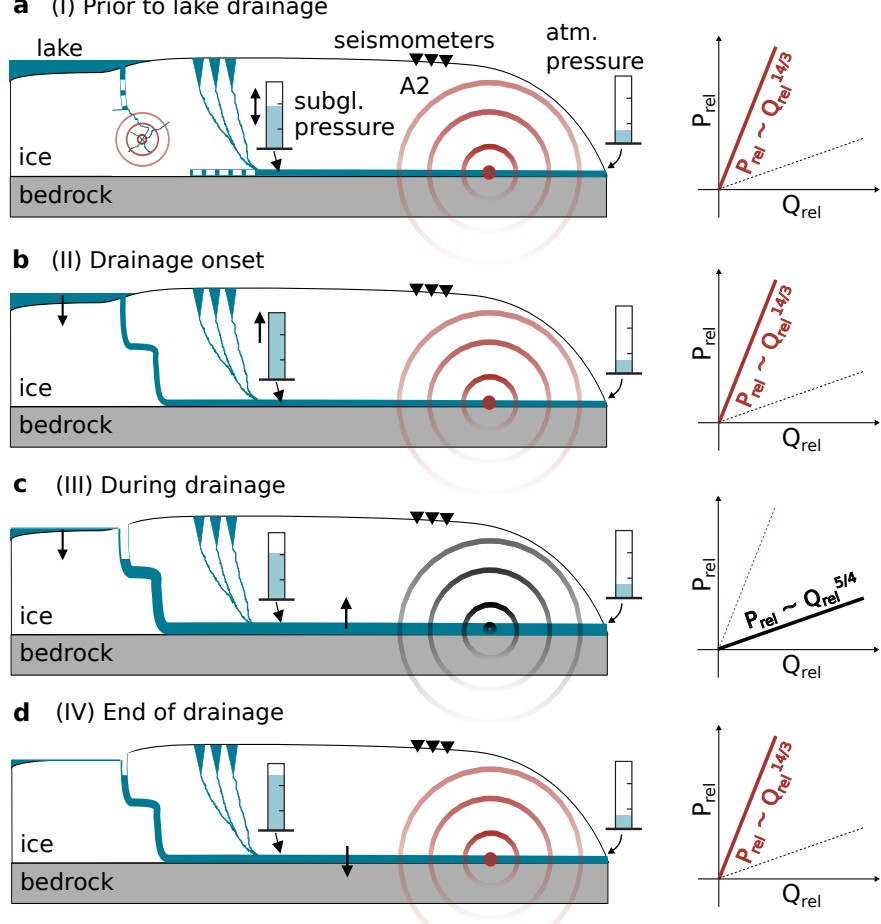

**Figure 13.** Illustration of the inferred history of subglacial hydraulics associated with the lake drainage. Shown is a schematic section along the major branch of the drainage system shown in Fig. 9. Blue indicates water, red and black circles seismic wave propagation which indicate discharge routing dominated by hydraulic pressure gradient adjustments and hydraulic radius adjustments, respectively. The dominant drainage regime after Gimbert et al. (2016) is also given on the right hand side. The times of the snapshots (I-IV) are indicated in Fig. 12. (a) Situation prior to the lake drainage. The lake reaches the drainage moulin which is not yet connected to the subglacial drainage system but icequake activity from the direction of the lake basin is increased (indicating hydrofracturing). Tremor generation beneath the glacier tongue is caused by the 'background' meltwater routing, and the pressure gradient measured between some arbitrary position along the subglacial conduit and the outlet (constant) is moderate but varies. (b) Initiation of the lake drainage. The drop in lake level causes an increase in the subglacial pressure gradient and local uplift of the ice. The capacity of the conduits is overwhelmed. (c) The subglacial conduits increase their radius by frictional melting to accommodate the lake discharge which results in a drop in the pressure gradient to pre-drainage values. (d) At the end of the lake drainage, as discharge decreases, the subglacial conduits shrink, causing a short episode of pressure gradient increase.