# Peer review of "Glaciohydraulic seismic tremors on an Alpine glacier"

_The Cryosphere, 2019_

## Referee Comment (RC1) · Anonymous Referee #1 · 26 Aug 2019

This is a very well written report on the analysis of a significant field effort to further develop cryo-seismological methods for the purpose of understanding glacier hydrology and processes related to hydrology. The data are unique and of high quality, and the analysis techniques are cutting edge. What speaks strongest for the positive regard of this manuscript is the creative way in which seismological methods and unusual seismic signals (as many are in this day and age, because cryoseismology is just starting as a field) are used to develop fundamental understanding about the glaciohydraulic system.

I did not find many problems or issues to comment on, as it was clear that the authors have done a very careful job of preparing their manuscript in advance of submitting it. My relatively minor comments are provided below.

[Figure]

page 2 line 10: rewrite this sentence to read something like "... these approaches have drawbacks, including: being expensive and laborious, providing subsurface images at only a few instances in time, and being isolated in location.

page 2 line 19, I would "call out" (i) and (ii) as two indented short sentences in a list, then put the sentences that discuss them, e.g., "This configuration is expected..." which begins on on line 20 in the rest of the paragraph that follows, and start the discussion sentences with "In the case of (i), this configuration is expected... In the case of (ii), These scaling relations..." etc. The way it reads now, the reader might not see the list of two end members in a simple way, because there is discussion involved in the definition of the list.

page 3 around line 24 - can a description of how the lake levels were monitored and how draining was detected be added? Was this done using instrumentation (e.g., depth or pressure sensors in the water) or was it done via remote sensing? How accurately or frequently are water depth measurements made? Also, since power outages are referred to, at the end of the page, the type of power source (photovoltaic?) should be mentioned. On the next page in the paragraph starting on line 5, it might be worth mentioning the sample rate of the GPS units and also the sample rates of the various data sets associated with Lenk and Geopravent...

page 5 around line 20 - out of curiosity, do the Rayleigh wave polarizations conform roughly with direction of radiation and source location? Would polarizations be capable, in the absence of other analysis, of determining source location or at least source azimuth? How consistent would location and azimuth be if just Rayleigh wave polarization were used to determine phase velocity vectors?

equation 4: is it necessary to have two sets of absolute value or "norm" ($|.|$) signs, one on the denominator and one on the whole fraction of the right hand side?

page 9 line 27. I seem to have forgotten what a "Bartlett processor" refers to. Was this defined above? Maybe make a citation to equation 3 instead of just referring to the

Bartlett processor. . ... Ditto with the term MVDR-Rayleigh results (a simple parenthetical with a equation number reference would be enough).

page 10 line 7 - Can "spurious body wave contributions" be defined more precisely? Are they whole-ice-thickness modes of P or S where the wave vector is horizontal?

page 10 line 19 - has the term MVDR-grids been defined?

---

## Referee Comment (RC2) · Alex Brisbourne (Referee) · 20 Sep 2019

General comments

Lindner et al. present a careful and thorough analysis of what looks to be a hard-won data set from a Swiss alpine glacier. The paper presents the application of a number of established techniques to investigate the relationship between glacial hydrology and seismicity / seismic tremor. The authors present some of the most convincing arguments to date supporting the applicability of cryo-seismology, with sufficient and appropriate data, to help further our understanding of glacial hydrology.

The paper is well structured and well written. It is clear that the authors have put a lot of work into a thorough analysis of the available data sets. The figures contain a wealth

of information that is, in general, presented clearly and succinctly. As such, I only have minor comments.

Specific comments

There is no discussion of the uncertainties associated with Bartlett and MVDR locations. Given the methodology and the tight array aperture, these could be significant. However, there is a reasonable degree of clustering and results are comparable and as such appear reliable. There certainly appears to be a statistical significance to the results. It does not appear that the authors have over-interpreted the results by ignoring uncertainties. However, for reference, and certainly for workers building on this study, it would be useful to quantify these uncertainties and discuss in context.

When measuring seismicity rates with an STA/LTA, is it necessary to account for variation in background noise? Could the masking of events during periods of high background noise lead to a reduced measured seismicity rate and vice versa?

I recall that Lennartz LE3D are flat from 1-80Hz but may be wrong (P3L26)

Technical corrections

The manuscript would benefit from a careful reading to identify a number of grammatical errors and referencing format issues. I highlight a number of minor issues below.

P2L37 - why e.g.?

P3L9 – what does "latter" refer to?

P4L7 (and elsewhere) – Simme River (capitalise river, valley etc)

P4L18 – blue curve in Fig. 2a and 2b (the latter is actually more instructive)

P4L20 – Is it possible to highlight the elevation of the moulin inlet on Fig. 2b?

P5L24 + P7L12 - "the the"

P6L22 – reword "allows to measure"

P8L24 – even not event

P9L4 - 7c not 7b

P9L9-10 – this sentence does not make sense.

P9L20 – remove "with"

P10L20 onwards – be sparing and consistent with parentheses around figure numbers (Fig, 8, left column)

P11L3 – "shown upstream area" – what does this mean?

P12L32 – "in the case where"

P13L22 – reformat Gimbert reference.

P13L27 – reword "more targeted as without"

P15L14 – reformat reference

P15L20 – "In the following" seems strange here.

P15L26 – (less than 100 m)

P15L30 – remove "also"

Figures

Fig. 2 – Is the stripe down the middle of 2e a data gap? If so, make white. There is no caption of e or f.

Fig. 11b it is hard to see the trend in the black dots. Can these be plotted on top?

---

## Author Comment (AC1) · 30 Oct 2019

**Author comments to RC1 (anonymous reviewer)**

Dear Reviewer, many thanks for your comments. In the following, we provide the corresponding author comments in a point-by-point style.

- page 2 line 10: rewrite this sentence to read something like "...these approaches have drawbacks, including: being expensive and laborious, providing subsurface images at only a few instances in time, and being isolated in location.

→ Thanks for the suggestion, we will implement these changes to improve the text flow.

[Figure]

- page 2 line 19, I would "call out" (i) and (ii) as two indented short sentences in a list, then put the sentences that discuss them, e.g., "This configuration is expected..."which begins on on line 20 in the rest of the paragraph that follows, and start the discussion sentences with "In the case of (i), this configuration is expected...In the case of (ii), These scaling relations..." etc. The way it reads now, the reader might not see the list of two end members in a simple way, because there is discussion involved in the definition of the list.

→ We will also work on this issue.

- page 3 around line 24 - can a description of how the lake levels were monitored and how draining was detected be added? Was this done using instrumentation (e.g., depth or pressure sensors in the water) or was it done via remote sensing? How accurately or frequently are water depth measurements made? Also, since power outages are referred to, at the end of the page, the type of power source (photovoltaic?) should be mentioned. On the next page in the paragraph starting on line 5, it might be worth mentioning the sample rate of the GPS units and also the sample rates of the various data sets associated with Lenk and Geopravent...

→ The lake level was monitored with a pressure sensor as indicated in Fig. 1. However, we will add information to answer the above questions since it does not seem to be clear from the current version.

- page 5 around line 20 - out of curiosity, do the Rayleigh wave polarizations conform roughly with direction of radiation and source location? Would polarizations be capa-ble, in the absence of other analysis, of determining source location or at least source azimuth? How consistent would location and azimuth be if just Rayleigh wave polarization were used to determine phase velocity vectors?

→ In principle, source azimuths can be determined from polarization analysis, as was e.g. done by Vore et al. (2018) for glaciohydraulic tremors. However, since we had arrays installed allowing for more accurate back azimuth / source locations, we didn't explore this option. A quick look into the polarization attributes revealed quite noisy data and showed that some more work would be required to obtain robust back azimuth measurements. But as stated by Koper Hawley (2010), the back azimuth $\theta_H$ in the polarization code we used is not well defined for strongly elliptical particle motion. We therefore refrain from a more detailed analysis of our polarization results.

Vore, M. E., Bartholomaus, T. C., Winberry, J. P., Walter, J. I., Amundson, J. M. (2019). Seismic Tremor Reveals Spatial Organization and Temporal Changes of Subglacial Water System. Journal of Geophysical Research: Earth Surface, 124(2), 427-446.

Koper, K. D., Hawley, V. L. (2010). Frequency dependent polarization analysis of ambient seismic noise recorded at a broadband seismometer in the central United States. Earthquake Science, 23(5), 439-447.

• equation 4: is it necessary to have two sets of absolute value or "norm" (|.|) signs, one on the denominator and one on the whole fraction of the right hand side?

→ Thanks for pointing this out, one is enough.

• page 9 line 27. I seem to have forgotten what a "Bartlett processor" refers to. Was this defined above? Maybe make a citation to equation 3 instead of just referring to the Bartlett processor..... Ditto with the term MVDR-Rayleigh results (a simple parenthet-ical with a equation number reference would be enough).

→ We will consider this suggestion in order to improve the text.

• page 10 line 7 - Can "spurious body wave contributions" be defined more precisely? Are they whole-ice-thickness modes of P or S where the wave vector is horizontal?

→ Here, we intend to say that the measured seismic signal is not purely composed of Rayleigh waves but may also contain some body wave energy. The source of these body waves is unknown and might be turbulent water flow as well as anthropogenic activity.

- page 10 line 19 - has the term MVDR-grids been defined?

→ This simply refers to the spatial grid used in the MVDR beamforming. We will rephrase this expression.

---

## Author Comment (AC2) · 30 Oct 2019

**Author Response to RC2 (Alex Brisbourne)**

Dear Reviewer, many thanks for your comments. In the following, we provide the corresponding author comments in a point-by-point style.

- There is no discussion of the uncertainties associated with Bartlett and MVDR locations. Given the methodology and the tight array aperture, these could be significant. However, there is a reasonable degree of clustering and results are comparable and as such appear reliable. There certainly appears to be a statistical significance to the results. It does not appear that the authors have over-interpreted the results by ignoring uncertainties. However, for reference, and

certainly for workers building on this study, it would be useful to quantify these uncertainties and discuss in context.

→ Thank you for bringing up this point. To our knowledge, there is no standard way or best practice of quantifying these uncertainties. One way would be to measure the width of the maxima, e.g. at 95% of the maximum value as is done in Eibl et al. (2017). Still, such uncertainty measurements are not comparable for Bartlett and MVDR processor or for different frequencies. In addition, results from matched-field processing which is used in our study strongly differ for cases where the source is located in the array ("point source") or outside the array (smeared lobe). A solution could be to show the contour lines of the Bartlett/MVDR values which would at least give a sense of the spatial uncertainties between subsequent measurements. Nevertheless, we would be very open for suggestions on other approaches.

Eibl, E. P. 7 others (2017). Tremor-rich shallow dyke formation followed by silent magma flow at Bárdarbunga in Iceland. Nature Geoscience, 10(4), 299.

- When measuring seismicity rates with an STA/LTA, is it necessary to account for variation in background noise? Could the masking of events during periods of high background noise lead to a reduced measured seismicity rate and vice versa?

→ As shown by Walter et al. (2008), the STA/LTA trigger sensitivity can indeed be influenced by variations in the background noise. However, even though our icequake detections could be biased, we do not see restrictions on our conclusions for the following reasons. (i) We show that high icequake rates, independent whether under- or overestimated, may affect the tremor amplitude. (ii) We focus on times with high discharge, which are expected to lower the trigger sensitivity. (iii) Similarly, our main observation – icequakes from the lake direction just prior to the drainage – falls in the daytime of a warm day with pronounced melt cycle

which is expected to decrease the trigger sensitivity. Additionally, no sustained icequake occurrence from this direction is observed beforehand where different background levels are observed.

Walter, F., Deichmann, N., Funk, M. (2008). Basal icequakes during changing subglacial water pressures beneath Gornergletscher, Switzerland. Journal of Glaciology, 54(186), 511-521.

- I recall that Lennartz LE3D are flat from 1-80Hz but may be wrong (P3L26)

→ According to the Lennartz specifications listed on their webpage, the response is flat up to 100 Hz. Could it be that an earlier sensor version was flat from 1-80Hz?

- Technical corrections
  The manuscript would benefit from a careful reading to identify a number of grammatical errors and referencing format issues. I highlight a number of minor issues below. (22 comments)

→ Many thanks for carefully reading the manuscript. We will implement these comments.

---

## Author Response (AR1)

Eidgenössische Technische Hochschule Zürich
Swiss Federal Institute of Technology Zurich

**Laboratory of**
**Hydraulics, Hydrology and Glaciology**

ETH Zurich
Fabian Lindner
PhD Student
HIA C 54.2
Hönggerbergring 26
8093 Zurich, Switzerland

Phone  +41 44 632 66 31
lindner@vaw.baug.ethz.ch

To: The editor of The Cryosphere

Zurich, November 22, 2019

**Revision of manuscript tc-2019-155**

Dear editor and reviewers,

We thank you for carefully reading our manuscript titled "Glaciohydraulic seismic tremors on an Alpine glacier". On the following pages, we provide a point-by-point response to the reviewers comments. Our replies are colored blueish and contain line numbers (where applicable) which refer to the revised manuscript. Comments which were straightforward to implement (such as typos) are simply ticked off (using the ✓ sign) without providing a response. In addition, enclosed you will also find a list of changes as well as an annotated PDF with all changes highlighted.

If you have any questions, we would be happy to answer them. We are looking forward to hearing from you about your decision.

Best regards,

Fabian Lindner

Comments of reviewer 1

This is a very well written report on the analysis of a significant field effort to further develop cryo-seismological methods for the purpose of understanding glacier hydrology and processes related to hydrology. The data are unique and of high quality, and the analysis techniques are cutting edge. What speaks strongest for the positive regard of this manuscript is the creative way in which seismological methods and unusual seismic signals (as many are in this day and age, because cryoseismology is just starting as a field) are used to develop fundamental understanding about the glaciohydraulic system.

I did not find many problems or issues to comment on, as it was clear that the authors have done a very careful job of preparing their manuscript in advance of submitting it. My relatively minor comments are provided below.

> page 2 line 10: rewrite this sentence to read something like "...these approaches have drawbacks, including: being expensive and laborious, providing subsurface images at only a few instances in time, and being isolated in location.
> Thanks for the suggestion, we adapted the text.

✓ page 2 line 19, I would "call out" (i) and (ii) as two indented short sentences in a list, then put the sentences that discuss them, e.g., "This configuration is expected... " which begins on on line 20 in the rest of the paragraph that follows, and start the discussion sentences with "In the case of (i), this configuration is expected...In the case of (ii), These scaling relations... " etc. The way it reads now, the reader might not see the list of two end members in a simple way, because there is discussion involved in the definition of the list.

> page 3 around line 24 - can a description of how the lake levels were monitored and how draining was detected be added? Was this done using instrumentation (e.g., depth or pressure sensors in the water) or was it done via remote sensing? How accurately or frequently are water depth measurements made? Also, since power outages are referred to, at the end of the page, the type of power source (photovoltaic?) should be mentioned. On the next page in the paragraph starting on line 5, it might be worth mentioning the sample rate of the GPS units and also the sample rates of the various data sets associated with Lenk and Geopravent...
> The lake level was monitored with a pressure sensor as indicated in Fig. 1. We added some information on the measurements and their temporal sampling (**P3L25-P4L13**).

> page 5 around line 20 - out of curiosity, do the Rayleigh wave polarizations conform roughly with direction of radiation and source location? Would polarizations be capable, in the absence of other analysis, of determining source location or at least source azimuth? How consistent would location and azimuth be if just Rayleigh wave polarization were used to determine phase velocity vectors?
> In principle, source azimuths can be determined from polarization analysis, as was e.g. done by Vore et al. (2018) for glaciohydraulic tremors. However, since we had arrays installed allowing for more accurate back azimuth / source locations, we didn't explore this option. A look into the polarization attributes revealed quite noisy data and showed that some more work would be required to obtain robust back azimuth measurements. But as stated by Koper & Hawley (2010), the back azimuth $\theta_H$ in the polarization code we used is not well defined for strongly elliptical particle motion. We therefore refrain from a more detailed analysis of our polarization results.
> Vore, M. E., Bartholomaus, T. C., Winberry, J. P., Walter, J. I., & Amundson, J. M. (2019). Seismic Tremor Reveals Spatial Organization and Temporal Changes of Subglacial Water System. Journal of Geophysical Research: Earth Surface, 124(2), 427-446.
> Koper, K. D., & Hawley, V. L. (2010). Frequency dependent polarization analysis of ambient seismic noise recorded at a broadband seismometer in the central United States. Earthquake Science, 23(5), 439-447.

equation 4: is it necessary to have two sets of absolute value or "norm" (|.|) signs, one on the denominator and one on the whole fraction of the right hand side?
Thanks for pointing this out, one is enough, we corrected this issue.

page 9 line 27. I seem to have forgotten what a "Bartlett processor" refers to. Was this defined above? Maybe make a citation to equation 3 instead of just referring to the Bartlett processor..... Ditto with the term MVDR-Rayleigh results (a simple parenthetical with a equation number reference would be enough).
We added a reference to the equations defining Bartlett and MVDR processor.

page 10 line 7 - Can "spurious body wave contributions" be defined more precisely? Are they whole-ice-thickness modes of P or S where the wave vector is horizontal?
Here, we intend to say that the measured seismic signal is not purely composed of Rayleigh waves but may also contain some body wave energy. The source of these body waves is unknown and might be turbulent water flow as well as anthropogenic activity. We removed the word "spurious" and hope this clarifies the issue.

page 10 line 19 - has the term MVDR-grids been defined?
This simply refers to the spatial grid used in the MVDR beamforming. We rephrased this expression.

Comments of reviewer 2 - Alex Brisbourne

General comments

Lindner et al. present a careful and thorough analysis of what looks to be a hard-won data set from a Swiss alpine glacier. The paper presents the application of a number of established techniques to investigate the relationship between glacial hydrology and seismicity / seismic tremor. The authors present some of the most convincing arguments to date supporting the applicability of cryo-seismology, with sufficient and appropriate data, to help further our understanding of glacial hydrology.

The paper is well structured and well written. It is clear that the authors have put a lot of work into a thorough analysis of the available data sets. The figures contain a wealth of information that is, in general, presented clearly and succinctly. As such, I only have minor comments.

Specific comments

There is no discussion of the uncertainties associated with Bartlett and MVDR locations. Given the methodology and the tight array aperture, these could be significant. However, there is a reasonable degree of clustering and results are comparable and as such appear reliable. There certainly appears to be a statistical significance to the results. It does not appear that the authors have over-interpreted the results by ignoring uncertainties. However, for reference, and certainly for workers building on this study, it would be useful to quantify these uncertainties and discuss in context.

Thank you for bringing up this point. To our knowledge, there is no standard way or best practice of quantifying these uncertainties. Contrary to our author comments, we decided to asses MFP uncertainties by investigating the determined source locations as a function of phase velocity as this is the only "tunable" parameter in our MFP approach. By doing so, in essence, we quantify the source location uncertainty due to neglecting lateral variations in phase velocity (which are known the be present from previous studies). We added a plot (**P21**) and a discussion to the Appendix (**P16L27-P17L16**) and a reference to it in the main text (**P10L14**). The results strengthen our results in the sense that the source locations discussed throughout the manuscript appear to be robust. In addition, we obtained very similar conclusions from a consideration of the source "smearing" by displaying the spatial extent of the area being within 95% of the MFP output maxima. It shows that A0 and A2 locations are better constrained (since the corresponding sources are close to the arrays) than A1 and A3 locations, where epicentral distances are not resolvable. We are aware, that other factors may influence the source location uncertainties (neglecting 3D effects, noise which decreases the coherence across the array), but we think that our analysis adequately addresses the uncertainties.

When measuring seismicity rates with an STA/LTA, is it necessary to account for variation in background noise? Could the masking of events during periods of high background noise lead to a reduced measured seismicity rate and vice versa?

As shown by Walter et al. (2008), the STA/LTA trigger sensitivity can indeed be influenced by variations in the background noise. However, even though our icequake detections could be biased, we do not see restrictions on our conclusions for the following reasons. (i) We show that high icequake rates, independent whether under- or overestimated, may affect the tremor amplitude. (ii) We focus on times with high discharge, which are expected to lower the trigger sensitivity. (iii) Similarly, our main observation – icequakes from the lake direction just prior to the drainage – falls in the daytime of a warm day with pronounced melt cycle which is expected to decrease the trigger sensitivity. Additionally, no sustained icequake occurrence from this direction is observed beforehand where different background levels are observed. We added this reasoning to the corresponding section (**P7L1-3**).

Walter, F., Deichmann, N., & Funk, M. (2008). Basal icequakes during changing subglacial water pressures beneath Gornergletscher, Switzerland. Journal of Glaciology, 54(186), 511-521.

I recall that Lennartz LE3D are flat from 1-80Hz but may be wrong (P3L26)
According to the Lennartz specifications listed on their webpage, the response is flat up to 100 Hz. Could it be that an earlier sensor version was flat from 1-80Hz?

Technical corrections
The manuscript would benefit from a careful reading to identify a number of grammatical errors and referencing format issues. I highlight a number of minor issues below.

P2L37 - why e.g.?
We removed the e.g.

P3L9 – what does "latter" refer to?
We rephrased this sentence.

✓ P4L7 (and elsewhere) – Simme River (capitalise river, valley etc)

P4L18 – blue curve in Fig. 2a and 2b (the latter is actually more instructive)
We added this.

P4L20 – Is it possible to highlight the elevation of the moulin inlet on Fig. 2B?
Unfortunately this is not possible, as we did not measure it. However, as described in the text, the lake "covered" the moulin inlet with a few meters of water column, before the moulin suddenly established a hydraulic connection such that water could spontaneously drain.

✓ P5L24 + P7L12 - "the the"

✓ P6L22 – reword "allows to measure"

P8L24 – even not event
We cannot see the what this comment refers to. Maybe the page or line number is wrong?

✓ P9L4 - 7c not 7b

P9L9-10 – this sentence does not make sense.
We rephrased this sentence.

✓ P9L20 – remove "with"

✓ P10L20 onwards – be sparing and consistent with parentheses around figure numbers (Fig, 8, left column)

P11L3 – "shown upstream area" – what does this mean?
The upstream area distributions (Fig. 9) show "the sum of all gridcell areas that are upstream and connected" (Flowers & Clarke, 1999) for each grid cell. We added some explanation in the manuscript to make it better understandable.
Flowers, G. E., & Clarke, G. K. (1999). Surface and bed topography of Trapridge Glacier, Yukon Territory, Canada: digital elevation models and derived hydraulic geometry. Journal of Glaciology, 45(149), 165-174.

✓ P12L32 – "in the case where"

P13L22 – reformat Gimbert reference.
We deleted this reference since it was repeating anyway.

✓ P13L27 – reword "more targeted as without"

P15L14 – reformat reference
There is no reference in that line. Maybe it is the wrong line number?

P15L20 – "In the following" seems strange here.
We removed "In the following" here.

✓ P15L26 – (less than 100 m)

✓ P15L30 – remove "also"

Figures

Fig. 2 – Is the stripe down the middle of 2e a data gap? If so, make white. There is no caption of e or f.
Yes, the yellow stripe in 2e is due to station maintenance and thus a data gap. We made the stripe white and corrected the caption.

Fig. 11b it is hard to see the trend in the black dots. Can these be plotted on top?
If we plot the black dots on top, the trend in the purple dots is hard to see. Since the black dots correspond to the times in the beginning of the time series which is not discussed in the manuscript, we feel that it is more appropriate to show the purple dots.

List of changes

- We added detail on the lake level measurement and technical detail on some instruments (sampling rates, power supply).
- We added a section to the appendix discussing the uncertainties of the source locations determined with matched-field processing. The section is supported by a new plot.
- We added a short discussion (two sentences) on the STA/LTA trigger sensitivity, which could potentially bias our icequake detection rates.
- We slightly adapted Figure 2e (replaced the vertical yellow stripe with a white one) to properly account for the data gap. Also, we adapted the caption of Figure 2.
- We implemented the reviewers' suggestions to improve the text.

[revised manuscript text omitted]
. Power supply for these stations was achieved via alkaline batteries which needed to be replaced on a weekly basis. In the following, consistent with the station names, we refer to our four arrays as A0 (stations PM01-PM06), A1 (PM11-PM15), A2 (PM21-PM25), and A3 (PM31-PM35). While A0 recordings are continuous (apart from gaps due to station maintenance), recordings from the other arrays suffer from occasional power outages and frequently exhibit gaps over midnight of up to 26 minutes. A0, A1, and A2 stations recorded data through early September (day 250 to 252, respectively), A3 stations were dismantled on August 23 (day 236) due to a slushy snow layer at the glacier's surface.

In addition to the seismogenic ground motion, we surveyed the (low-frequency) glacier surface motion due to e.g. ice flow and glacier hydraulics at three locations using GPS units (Fig. 1a) (two-hour sampling interval after post-processing). Furthermore, we make use of the following time series: discharge in the Simme  River to the north (measured ≈4  km from the terminus of Rätzligletscher), level of the outlet stream (≈1.5  km from the terminus of Rätzligletscher), and level of Lac des Faverges. Simme discharge is  provided as hourly averages by Switzerland's Federal Office for the Environment, and the stream and lake level are provided through a monitoring program conducted by the municipality of Lenk and the company Geopraevent. The lake level was monitored by a pressure sensor (Fig. 1) sampling data at ten minute intervals.

[revised manuscript text omitted]

**4.2.3 Array A2**

A2 shows less variation than A1 and the source clustering suggests a close tremor source to the northwest of the array in the direction of the small glacier tongue (A2-1). During and after the lake drainage and similar to A1, this source seems to wander slightly farther away towards the north (back azimuth increase of 5-10°). Again, however, epicentral distance is not

well resolved. In some instances and mainly concurrent with discharge peaks, additional signals arrive from a more distant source west of the array (A2-2).

**4.2.4 Array A3**

Tremor signals observed at A3 mainly arrive at the array from the west with back azimuths in the range of approximately 250-285° (A3-1). The epicentral distances of these sources cannot be constrained but their back azimuth points towards a region of the glacier where several moulins are located. Some of them have a sinkhole-like structure  tens of meters in diameter and are stationary over decades (Huss et al., 2013; GLAMOS, 2018). We also note that a military radar facility is located at a back azimuth of approximately 275° and a distance of around 2  km from the array center, whose operation cannot be excluded as a noise source for A3. Another tremor source is located southwest at a back azimuth of around 230° (A3-2). This source clusters closer than twice the array aperture, is collocated with the position of a moulin identified from ortophotographs, and appears to be active during peak discharges.

**4.2.5 Discussion**

We also test the MVDR results for plausibility by comparing them to the solutions obtained by using the Bartlett processor (eq. 3). Even though these results were obtained for a smaller spatial grid in order to save computation time, both processors yield similar results. In addition, we also test the robustness of our results by (apart from testing a grid of coordinates) allowing also a grid search over phase velocity from 1500  ms$^{-1}$ to 3500  ms$^{-1}$ in 50  ms$^{-1}$ steps. Compared to the MVDR-Rayleigh results (eq. 4), both the back azimuth and the epicentral distance scatter more broadly but the general source distribution stays similar. The velocities for which the Bartlett results are maximized are systematically higher than the Rayleigh wave velocities used previously, especially for A2 (median of approximately 2800  ms$^{-1}$ versus 2200  ms$^{-1}$ for Rayleigh waves). However, the average Bartlett maximum is increased only marginally (0.86 for Bartlett-Rayleigh MFP versus 0.88 for Bartlett MFP with velocity grid search), which indicates that there is a tradeoff between epicentral distance and velocity. Here, we note that Walter et al. (2015) find quickly growing uncertainties in epicentral distance estimates of icequakes with distance from the array center. These uncertainties in the source locations evident also in Fig. 7 are further discussed in Appendix A.

[revised manuscript text omitted]

As discussed in Sect. 4.2.5, MFP source location uncertainties, in particular epicentral distances, are considerable four sources outside the arrays. In our MFP formulation (Sect. 4.1), the synthetic wavefield used to match the field data is dependent on the source-receiver distances and the velocity model. The latter is assumed homogeneous and fixed to the phase velocity values reported in Lindner et al. (2018) at each array, thus neglecting lateral velocity variations. In addition, velocities that maximize the MFP output tend to be systematically increased for A2 (Sect. 4.2.5). To investigate the source location uncertainty caused by simplifying the velocity model, we consider the source locations for the two times shown in Fig. 8 as a function of phase velocity. To this end, we apply MVDR-MFP to a 10 m spatial grid and test phase velocities from $1500 \, \mathrm{ms}^{-1}$ to $2500 \, \mathrm{ms}^{-1}$ in $50 \, \mathrm{ms}^{-1}$ steps, which is the phase velocity range for frequencies of 8.5 to 12 Hz (Lindner et al., 2018). Fig. 21 shows that A0 locations cluster tightly (order of tens of meters) around value estimated for a constant velocity model and Rayleigh waves (blue plus-signs in Fig. 21) for both time intervals, which indicates robust source location estimates. The same holds for A2 source locations that, even though outside the array, are largely unaffected (few tens of meters) by phase velocity variations. This suggests a close-by tremor source and is further supported by the side lobe of the A3 MFP results, which points points to the same region from a different angle (Fig. 8). In contrast to A0 and A2, A1 and A3 epicentral distances strongly depend on the velocity model (source locations affected by hundreds of meters). Especially in the MFP example from 30 August (lower panel in Fig. 21), A1 cannot resolve the epicentral distances, indicated by the source location clustering at the edge of the spatial grid.

In addition to the simplified velocity model, we neglect surface topography and assume sources located at the surface. Especially for A2, increased velocities hint towards a body wave contribution, which could originate from a close-by channel at the glacier bed. This suggests, that source location uncertainties could be further affected by our two-dimensional MFP setup.

**Appendix B: Subglacial drainage**

Beneath glaciers, water flows in response to the hydraulic potential $\phi$, which is the sum of the pressure potential and the elevation potential (Shreve, 1972), i.e.

$$\phi = f \rho_i g h_i + \rho_w g z_b, \tag{B1}$$

where $f$ is the flotation fraction, $\rho_i = 910 \, \mathrm{kgm}^{-3}$ and $\rho_w = 1000 \, \mathrm{kgm}^{-3}$ are the densities of ice and water, $g = 9.81 \, \mathrm{ms}^{-2}$ is the gravitational acceleration, $h_i$ is the (laterally varying) ice thickness and $z_b$ is the bedrock elevation. Measurements of the ice thickness are available along a grid of flight profiles where the glacier bed was surveyed with helicopter-borne ground-penetrating radar (GPR, Langhammer et al., 2018; Grab et al., 2018). We interpolate the ice thickness values available along the GPR-profiles to a regular 50-m-grid using inverse distance weighting (Shepard, 1968) of the 100 nearest data points and their corresponding ice thicknesses. In addition to the GPR profiles, we also use the coordinates of

5 the glacier margin (e.g. Fig. 1) for the interpolation, where we set the ice thickness to zero. We then calculate the bedrock topography by subtracting the ice thickness from the digital elevation model. Subsequently, we calculate the hydraulic potential for $f = 1.0$ (water pressure equals the ice-overburden pressure), since we expect high water pressures, especially during the lake drainage initiation (Roberts, 2005). This is confirmed by continuous GPS measurements in the vicinity of A0, A2, and A3, which show vertical lifting during the first $\approx$8-36 hours of the lake drainage (Fig. 2b). In addition, we also consider the

10 hydraulic potential calculated for (spatially uniform) water pressures of half the ice overburden for comparison.

To investigate likely subglacial water-flow paths, we calculate the upstream area for each grid cell, i.e. the (grid cell) area that is upstream and connected to the grid cell of consideration. We follow the approach of (Flowers and Clarke, 1999) and calculate the upstream area distribution using the Quinn algorithm (Quinn et al., 1991), which transfers the area to all downstream cells among the eight direct neighbor cells weighted by the relative gradients. We perform depression filling of the hydraulic potential

15 surfaces and subsequent calculation of the upstream area using the RichDEM toolbox (Barnes, 2016). While the results might suffer from inaccuracies introduced by the interpolation of the ice thickness profiles and by neglecting (horizontal) englacial transport as well as subglacial mechanics (Flowers and Clarke, 1999), they are consistent with field observations (see main text for details).

**Appendix C: Tremor-discharge scaling**

20 Discharge data is provided in hourly averages, while tremor amplitude samples are calculated from 30 minutes of data with 50 percent overlap, resulting in a sample spacing of 15 minutes. For consistency and to smooth the (partly) noisy tremor data (Fig. 4(b)), we also calculate running averages of the tremor amplitude by taking a window of five samples centered around each timestamp associated with the discharge data. In addition, we test corrections of the discharge time series for the time it takes the water from the glacier terminus to the gauging station ($\approx$4.5  km horizontal distance and $\approx$1.5  km elevation

25 difference) by up to two hours travel time but this does not change our conclusions.

*Author contributions.* FL, FW, and GL designed the experiments, which were carried out by all authors. FL processed and analyzed the data with the help of FW and FG. FL prepared the manuscript with contributions from all co-authors.

*Competing interests.* The authors declare that they have no conflict of interest.

*Acknowledgements.* This work was funded by the Swiss National Science Foundation project Glacial Hazard Monitoring with Seismology (GlaHMSeis project PP00P2 157551). The geophones and DataCubes for 15 stations were provided by the Geophysical Instrument Pool Potsdam (GIPP) under project AnICEotropy. We thank Andreas Bauder who installed the GPS stations and Philippe Limpach who processed the GPS data, as well as Lorenz Meier from Geopraevent for sharing the data from the lake monitoring. We are also grateful to our technicians Pascal Graf and Christian Scherrer, as well as to Adrian Doran and all other people who helped in the field. We appreciate the collaboration with the municipality of Lenk and would like to acknowledge logistical support from Remontées Mécaniques de Crans-Montana (CMA), the Swiss Armed Forces, and the Federal Office of Civil Aviation. We used ObsPy (Beyreuther et al., 2010) for seismic data processing and created the figures with the Matplotlib plotting library for Python (Hunter, 2007). We acknowledge the constructive comments from the editor Jürg Schweizer, Alex Brisbourne and another anonymous reviewer.

[revised manuscript text omitted]

**Figure 1.** (a) Map of the extent of Glacier de la Plaine Morte (thick black line), topography (contour lines and color-coding), and location of sensor installations (symbols). Seismic stations are numbered for each array (A0-A3) counterclockwise from 1 (north; northeast for A0) to 5 (center station). Station PM06 (lower center station of A0) was added end of July. The blue shaded area depicts the approximate maximum extent of Lac des Faverges with the moulin through which the drainage initiated (black cross). The arrows indicate the direction and distance to the discharge gauge of the Simme  River and to the weather stations ABO and MVE. (b) Sentinel-2 imagery (modified Copernicus Sentinel data 2019/Sentinel Hub) acquired on 2016-08-23 (day 236) with the glacier extent from (a). SL stands for supraglacial lake. (c) Orthophoto taken on 2016-09-07 (day 251) with the glacier extent from (a).

**Figures**

[Figure]

**Figure 2.** (a) Hydrological data recorded in the vicinity of Glacier de la Plaine Morte: discharge of the Simme  River measured in the Simme Valley (blue curve; ≈4  km line of sight from the glacier terminus), level of the Truebbach stream (gray curve; ≈1.5  km from glacier terminus), height of the water column in the lake above the pressure sensor (orange dashed), precipitation at stations ABO and MVE (blue and purple bars; 12  km north and 10  km south of the glacier, respectively). (b) Discharge and lake level for the drainage period (same as (a)) and the vertical displacement of three GPS units (black lines). (c) Spectrogram of station PM05 for the same time period shown in (a). (d)  Zoom-in of the spectrogram in (c) showing anthropogenic noise. (de) Zoom-in of a spectrogram of station PM32 showing moulin resonances. The white bar indicates a data gap due to station maintenance (f) Zoom-in of the spectrogram in (c) showing the lake drainage. 
[revised manuscript text omitted]

[Figure]

**Figure 21.** Source locations from MFP (MVDR processor) as a function of phase velocity (colored dots) to assess uncertainties. Results are shown for the two time windows also shown in Fig. 8. Size of the dots scales with the MVDR output values, blue plus-signs indicate the source locations using a homogeneous velocity model and Rayleigh wave velocities, and the white cross in the lower panel indicates the position of the main drainage moulin.